# The impact of resolution on meteorological, chemical and aerosol properties in regional simulations with WRF-Chem

P. Crippa[1], R. C. Sullivan[2], A. Thota[3], S. C. Pryor[2,3]

[1]COMET, School of Civil Engineering and Geosciences, Cassie Building, Newcastle University, Newcastle upon Tyne, NE1 7RU, UK

[2]Department of Earth and Atmospheric Sciences, Bradfield Hall, 306 Tower Road, Cornell University, Ithaca, NY 14853, USA

[3]Pervasive Technology Institute, Indiana University, Bloomington, IN 47405, USA

*Correspondence to:* P. Crippa (paola.crippa@ncl.ac.uk), School of Civil Engineering and Geosciences, Cassie Building, Room G15, Telephone: +44 (0)191 208 5041, Newcastle University, Newcastle upon Tyne, NE1 7RU, UK

**Abstract**
Limited area (regional) models applied at high resolution over specific regions of interest are
generally expected to more accurately capture the spatio-temporal variability of key
meteorological and climate parameters. However, improved performance is not inevitable, and
there remains a need to optimize use of numerical resources, and to quantify the impact on
simulation fidelity that derives from increased resolution. The application of regional models
for climate forcing assessment is currently limited by the lack of studies quantifying the
sensitivity to horizontal spatial resolution and the physical-dynamical-chemical schemes
driving the simulations. Here we investigate model skill in simulating meteorological, chemical
and aerosol properties as a function of spatial resolution, by applying the Weather Research
and Forecasting model with coupled Chemistry (WRF-Chem) over eastern North America at
different resolutions. Using Brier Skill Scores and other statistical metrics it is shown that
enhanced resolution (from 60 to 12 km) improves model performance for all of the
meteorological parameters and gas phase concentrations considered, in addition to both mean
and extreme Aerosol Optical Depth (AOD) in three wavelengths in the visible relative to
satellite observations, principally via increase of potential skill. Some of the enhanced model
performance for AOD appears to be attributable to improved simulation of meteorological
conditions (notably precipitation and near-surface specific humidity) and the concentration of
key aerosol precursor gases (e.g. $SO_2$ and $NH_3$).
**Keywords:** added value, high-resolution WRF-Chem simulations, precipitation, aerosol
optical properties, extreme AOD

## 1 Motivation and Objectives

Aerosols alter Earth's radiation balance primarily by scattering or absorbing incoming solar radiation (direct effect, dominated by accumulation mode (diameters ~ wavelength ($\lambda$), where total extinction is often quantified using AOD), or regulating cloud formation/properties by acting as cloud condensation nuclei (CCN) (indirect effect, dominated by diameters $\geq$ 100 nm, magnitude = $f$(composition)). Most aerosols (excluding black carbon) have a larger scattering cross-section than absorption cross-section, and act as CCN thus enhancing cloud albedo and lifetimes. Hence increased aerosol concentrations are generally (but not uniformly) associated with surface cooling (offsetting a fraction of greenhouse gas warming) (Boucher, 2013;Myhre et al., 2013b) to a degree that is principally dictated by the aerosol concentration, size and composition, in addition to the underlying surface and height of the aerosol layer (McComiskey et al., 2008). Despite major advances in measurement and modeling, both the current global mean aerosol direct effect (possible range: -0.77 to +0.23 W m$^{-2}$) and the indirect effect (possible range: -1.33 to -0.06 W m$^{-2}$) remain uncertain (Stocker, 2013), as does their future role in climate forcing (Rockel et al., 2008) and regional manifestations (Myhre et al., 2013a). Specific to our current study region (eastern N. America), one analysis using the NASA GISS global model found that the "regional radiative forcing from US anthropogenic aerosols elicits a strong regional climate response, cooling the central and eastern US by 0.5–1.0 °C on average during 1970–1990, with the strongest effects on maximum daytime temperatures in summer and autumn. Aerosol cooling reflects comparable contributions from direct and indirect radiative effects" (Leibensperger et al., 2012). A recent comparison of multiple global models conducted under the AEROCOM-project indicated this is also a region that exhibits very large model-to-model variability in simulated AOD (<AOD> ~ 0.5, $\sigma$(AOD) ~ 1) (Myhre et al., 2013a).

Major reasons why aerosol radiative forcing on both the global and regional scales remains uncertain include short atmospheric residence times and high spatio-temporal variability of aerosol populations, and the complexity of the processes that dictate aerosol concentrations, composition and size distributions (Seinfeld and Pandis, 2016). Although aerosol processes and properties are increasingly being treated in the global Earth System Models (ESMs) (Long et al., 2015;Tilmes et al., 2015) applied in the Coupled Model Intercomparison Project Phase 6 (CMIP-6) (Meehl et al., 2014), the scales on which such models are applied remain much coarser than those on which aerosol population properties are known to vary (Anderson et al., 2003). Therefore, limited area atmospheric models (regional models) applied at higher

resolution over specific regions of interest are expected to 'add value' (i.e. improve the fidelity)
of the physical-dynamical-chemical processes that induce extreme events and dictate climate
forcing. There is empirical evidence to suggest a strong resolution dependence in simulated
aerosol particle properties. For example, WRF-Chem simulations with spatial resolution
enhanced from 75 km to 3 km exhibited higher correlations and lower bias relative to
observations of aerosol optical properties over Mexico likely due to more accurate description
of emissions, meteorology and of the physicochemical processes that convert trace gases to
particles (Gustafson et al., 2011;Qian et al., 2010). This improvement in the simulation of
aerosol optical properties implies a reduction of the uncertainty in associated aerosol radiative
forcing (Gustafson et al., 2011). Further, WRF-Chem run over the United Kingdom and
Northern France at multiple resolutions in the range of 40-160 km, underestimated AOD by
10-16% and overestimated CCN by 18-36% relative to a high resolution run at 10 km, partly
as a result of scale dependence of the gas-phase chemistry and differences in the aerosol uptake
of water (Weigum et al., 2016).
However, debate remains regarding how to objectively evaluate model performance, quantify
the value added by enhanced resolution (Di Luca et al., 2015;Rockel et al., 2008) and on
possible limits to the improvement of climate representation in light of errors in the driving
"imperfect lateral boundary conditions" (Diaconescu and Laprise, 2013). Nevertheless,
although "it is unrealistic to expect a vast amount of added values since models already
performs rather decently" (Di Luca et al., 2015) and global ESMs are now run at much higher
resolution than in the past, it is generally assumed that high resolution regional models will add
value via more realistic representation of spatio-temporal variability than global coarser-
resolution simulations. Further, "the main added value of a regional climate model is provided
by its small scales and its skill to simulate extreme events, particularly for precipitation"
(Diaconescu and Laprise, 2013).
It is particularly challenging to assess the added-value from enhanced resolution in the context
of climate-relevant aerosol properties since they are a complex product of the fidelity of the
simulation of meteorological parameters, gas-phase precursors, emissions and the treatment of
aerosol dynamics. Here we quantify the value added by enhanced resolution in the description
of physical and chemical atmospheric conditions using year-long simulations from WRF-Chem
over eastern North America, and investigate how they impact AOD. The primary performance
evaluation of aerosol properties focuses on AOD at different wavelengths ($\lambda$ = 470, 550 and
660 nm, where the AOD at different $\lambda$ is used as a proxy of the aerosol size distribution (Tomasi
et al., 1983), see details in Sect. 2.3) and is measured relative to observations from satellite-
borne instrumentation. Thus the term "value-added" is used here in the context of columnar
aerosol properties to refer to an improvement of model performance in simulation of
wavelength specific AOD as measured by the MODerate resolution Imaging
Spectroradiometer (MODIS) instrument aboard the polar-orbiting Terra satellite. To attribute
sources of the enhanced fidelity of AOD, our analysis also incorporates evaluation of the value-
added by enhanced resolution in terms of key meteorological and gas-phase drivers of aerosol
concentrations and composition and is conducted relative to the MERRA-2 reanalysis product
for the physical variables and columnar gas concentrations from satellite observations (see
details of the precise data sets used given below). The meteorological parameters considered
are air temperature at 2 m ($T_{2m}$), total monthly precipitation (*PPT*), planetary boundary-layer
height (*PBLH*) and specific humidity in the boundary layer ($Q_{PBL}$). The gas phase
concentrations considered are sulfur dioxide ($SO_2$), ammonia ($NH_3$), nitrogen dioxide ($NO_2$)
and formaldehyde (HCHO).
We begin by quantifying the performance of WRF-Chem when applied over eastern North
America at a resolution of 60 km (WRF60) (~ finest resolution likely to be employed in CMIP-
6 global simulations) and then compare the results to those from simulations conducted at 12
km (WRF12) (simulation details are given in Table 1). Quantification of model skill is
undertaken by mapping the WRF12 output to the WRF60 grid (WRF12-remap) and computing
Brier Skill Scores (BSS) using MODIS as the target, WRF60 as the reference forecast and
WRF12-remap as the forecast to be evaluated. We also evaluate the performance of the WRF-
Chem simulations of 2008 relative to climatology as represented by MODIS observations for
2000-2014. We additionally assess the impact of simulation resolution on extreme AOD values
that are associated with enhanced impacts on climate and human health. This analysis uses both
*Accuracy* and *Hit Rate* as the performance metrics and focuses on the co-occurrence of extreme
values in space from the model output and MODIS.
Based on the performance evaluation of the WRF-Chem simulations that indicate substantial
dry bias in the WRF60 simulations and large seasonality in the skill-scores for AOD as a
function of resolution, we conducted two further year-long simulations at 60 km. In the first
we held all other simulation conditions constant but selected a different cumulus
parameterization. In the second, we held all simulation conditions constant but employed a
different set of lateral boundary conditions for the meteorology. In the context of the
precipitation biases reported herein it is worthy of note that discrepancies in simulated
precipitation regimes are key challenges in regional modelling (both physical and coupled with
chemistry). Although the Grell 3D scheme has been successfully applied in a number of prior
analysis wherein the model was applied at resolutions in the range of 1-36 km (e.g. (Grell and
Dévényi, 2002;Lowrey and Yang, 2008;Nasrollahi et al., 2012;Sun et al., 2014;Zhang et al.,
2016)), the North American Regional Climate Change Assessment Program (NARCCAP)
simulations with WRF at 50-km were also dry biased in the study domain (Mearns et al., 2012).
Although there have been a number of studies that have sought to evaluate different cumulus
schemes over different regions at different resolutions, no definitive recommendation has been
made regarding the dependence of model skill on resolution and cumulus parameterization
(Arakawa, 2004;Jankov et al., 2005;Nasrollahi et al., 2012;Li et al., 2014). Hence, further
research is needed to identify the optimal cumulus scheme for use over North America at
coarser resolution. Thus, we performed a sensitivity analysis on the cumulus scheme at 60 km
by applying the Grell-Freitas parameterization (Grell and Freitas, 2014), which is the next
generation of the Grell 3D scheme.
**2 Materials and Methods**
**2.1 WRF-Chem simulations**
WRF-Chem (version 3.6.1) simulations were performed for the calendar year 2008 over eastern
North America, in a domain centered over southern Indiana (86°W, 39°N) at two resolutions,
one close to the finest resolution designed for CMIP-6 global model runs (i.e. 60 km, WRF60)
and the other one at much higher resolution (12 km, WRF12). Simulation settings are identical
for the two runs except for the time-step used for the physics (Table 1). Physical and chemical
parameterizations were chosen to match previous work using WRF-Chem at 12 km on the same
region which showed good performance relative to observations and the year 2008 was selected
because it is representative of average climate and aerosol conditions during 2000 - 2014
(Crippa et al., 2016). More specifically the simulations adopted the RADM2 chemical
mechanism (Stockwell et al., 1990) and a modal representation of the aerosol size distribution
(MADE/SORGAM, (Ackermann et al., 1998;Schell et al., 2001)) with three lognormal modes
and fixed geometric standard deviations (i.e. 1.7, 2 and 2.5 for Aitken, accumulation and coarse
mode, respectively (Ackermann et al., 1998;Grell et al., 2005)). Aerosol direct feedback was
turned on and coupled to the Goddard shortwave scheme (Fast et al., 2006). A telescoping
vertical grid with 32 model layers from the surface to 50 hPa and 10 layers up to 800 hPa was
selected. Meteorological initial and boundary conditions from the North American Mesoscale
Model at 12 km resolution (NAM12) are applied every 6 hours, while initial and chemical
boundary conditions are taken from MOZART-4 (Model for Ozone and Related chemical
Tracers, version 4) with meteorology from NCEP/NCAR-reanalysis (Emmons et al., 2010).
Anthropogenic emissions are specified for both WRF60 and WRF12 from the US National
Emission Inventory 2005 (NEI-05) (US-EPA, 2009) which provides hourly point and area
emissions at 4 km on 19 vertical levels. The simulation settings and specifically the use of a
modal representation of the aerosol size distribution were selected to retain computational
tractability. Accordingly, the 60 km simulations for the year 2008 completed in 6.4 hours
whereas the 12 km simulations completed in 9.5 days (230 hours) on the Cray XE6/XK7
supercomputer (Big Red II) owned by Indiana University, using 256 processors distributed on
8 nodes.
As described in detail below, in the WRF60 simulations configured as described in Table 1,
simulated precipitation during the summer months exhibits substantial dry bias, and the
analysis of value added by enhanced simulation resolution exhibited strong seasonality. We
performed a sensitivity analysis to the cumulus scheme, by conducting an additional year-long
simulation at 60 km using the Grell-Freitas parameterization (Grell and Freitas, 2014), which
is an evolution of Grell 3D that is scale-aware and treats some aspects of aerosol-cloud
interactions. We also tested the sensitivity of the simulation results to the meteorological
boundary conditions, by repeating the WRF60 simulations using output from the Global
Forecast System (GFS) at 0.5° resolution every 6 hours to provide the lateral boundary
conditions.
**2.2 Observations**
Model aerosol optical properties are evaluated relative to the MODIS Collection 6 dark-target
land aerosol product from aboard the Terra satellite (~1030 overpass local solar time (LST))
(Levy et al., 2013). To provide a consistent assessment of model skill, the evaluation of AOD
is conducted only on land areas since the MODIS dark-target ocean aerosol product is based
on a retrieval algorithm different from the one over land (Levy et al., 2013). Trace gas
concentrations are evaluated relative to measurements from the Ozone Monitoring Instrument
(OMI; version 3) (Chance, 2002) and the Infrared Atmospheric Sounding Interferometer (IASI;
NN version 1) (Whitburn et al., 2016) aboard the Aura (~1345 LST) and MetOp satellites
(~0930 LST), respectively. MODIS retrieves AOD at multiple $\lambda$ including 470, 550, and 660
nm, and the MODIS algorithm removes cloud-contaminated pixels prior to spatial averaging

over $10 \times 10$ km (at nadir). OMI and IASI have nadir resolutions of $13 \times 24$ km and 12 km (circular footprint), respectively, and have been filtered to remove retrievals with cloud fractions > 0.3 (Fioletov et al., 2011;McLinden et al., 2014;Vinken et al., 2014) and OMI pixels affected by the row anomalies. MODIS, OMI, and IASI provide near daily global coverage, although the row anomalies render portions of the OMI viewing swath unusable. Uncertainty in AOD from MODIS is spatially and temporally variable. It has been estimated as $\pm$ (0.05 + 15%) for AOD over land (Levy et al., 2013), and prior research has reported 71% of MODIS Collection 5 retrievals fall within 0.05 ± 20% for AOD relative to AERONET in the study domain (Hyer et al., 2011). The accuracy of OMI ("root sum of the square of all errors, including forward model, inverse model, and instrument errors" (Brinksma et al., 2003)) is 1.1 DU or 50% for $SO_2$, $2 \times 10^{14}$ cm$^{-2}$/30% for background/polluted $NO_2$ conditions, and 35% for HCHO. This uncertainty is typically reduced by spatial and temporal averaging, as employed herein (Fioletov et al., 2011;Krotkov et al., 2008). IASI $NH_3$ retrievals do not use an a priori assumption of emissions, vertical distribution, or lifetime of $NH_3$ (i.e. no averaging kernel); therefore, $NH_3$ accuracy is variable (Whitburn et al., 2016), and thus only retrievals with uncertainty lower than the retrieved concentrations are used herein.

For the model evaluation, satellite observations for each day are regridded to the WRF-Chem discretization. This is done by averaging all valid retrievals within: 0.1° and 0.35° of the WRF-Chem grid-cell center for the 12×12 km and 60×60 km resolutions, respectively for MODIS; 0.125° × 0.18° (along-track/latitudinal × cross-track/longitudinal) and 0.365° × 0.42° for OMI; 0.12° and 0.36° for IASI. To avoid issues from under-sampling, we require at least 10 valid MODIS granules for the 60×60 km daily average to be computed and at least 5 daily averages to compute a monthly average for each grid cell. Model evaluation of gaseous species is performed on a seasonal basis using standard scores (z-scores), which are computed as the difference between the seasonal mean within a grid cell and the seasonal spatial mean, divided by the seasonal spatial standard deviation. Use of z-scores allows comparison of the spatial patterns of satellite observations and model output in terms of standard deviation units from the mean.

The simulated meteorological properties are evaluated using Modern-Era Retrospective analysis for Research and Applications (MERRA-2) reanalysis data as the target. MERRA-2 is a homogenized and continuous in time description of atmospheric properties on a 3-dimensional global grid (horizontal resolution of 0.5°×0.625°, L72), developed by NASA and

was released in Fall 2015 (Molod et al., 2015). MERRA-2 provides hourly values of $T_{2m}$ and
*PBLH*, and vertical profile of 3-dimensional variables every 3 hours on a large number of
pressure levels. Here we compute the total specific humidity ($Q_{PBL}$) of the lowest 8 pressure
levels (i.e. in the boundary-layer approximated as the layer from 1000 to 825 hPa) in MERRA-
2, assuming an average air density in the PBL of 1.1 kg m$^{-3}$. For the evaluation of simulated
precipitation we use accumulated monthly total values.

**2.3 Spectral dependence of AOD**

Three properties dictate the actual aerosol direct radiative forcing: AOD, single scattering
albedo and asymmetry factor, all of which are a function of the wavelength ($\lambda$) of incident
radiation. The first property is related to the total columnar mass loading, typically dominates
the variability of direct aerosol effect (Chin et al., 2009) and is the focus of the current research.
The relationship between the aerosol size distribution and spectral dependence of AOD is
described by a power law function:

$$b\left(\lambda_1\right) = b\left(\lambda_2\right) \times \left(\frac{\lambda_1}{\lambda_2}\right)^{-a} \quad \textbf{(1)}$$

where $\beta$ is the particle extinction coefficient at a specific wavelength $\lambda$, and $\alpha$ is the Ångström
exponent (Ångström, 1964) which describes the wavelength dependence of AOD (and is
inversely proportional to the average aerosol diameter):

$$a = \frac{\ln \dfrac{AOD\left(\lambda_1\right)}{AOD\left(\lambda_2\right)}}{\ln \dfrac{\lambda_2}{\lambda_1}} \quad \textbf{(2)}$$

The aerosol volume distribution usually conforms to a multi-lognormal function with *n* modes:

$$\frac{dV(r)}{d \ln r} = \sum_{i=1}^{n} \frac{C_i}{\sqrt{2\pi}\sigma_i} \exp\left[\frac{-\left(\ln r - \ln R_i\right)^2}{2\sigma_i^2}\right] \quad \textbf{(3)}$$

where *r* is the particle radius and $C_i$, $R_i$ and $\sigma_i$ are the particle volume concentration, the
geometric mean radius and the standard deviation in the mode *i* respectively.
We can thus compute AOD for a polydisperse distribution of aerosols with refractive index *m*
in an atmospheric column of height Z as:
$$AOD(\lambda) = \int \frac{3\beta(m,r,\lambda)}{4r} \frac{dV(r)}{d\ln r} d\ln r dZ \quad \textbf{(4)}$$
As indicated in (Schuster et al., 2006), "the spectral variability of extinction diminishes for
particles larger than the incident wavelength", thus fine mode particles contribute more to AOD
in the visible ($\lambda\sim0.5$ µm) than at longer wavelengths, whereas coarse mode particles provide a
similar AOD both at short and long wavelengths. This is reflected in the Ångström parameter
which can be thus used as a proxy for the fine mode fraction or fine mode radius (Schuster et
al., 2006).
**2.4 Quantification of model performance and added-value**
Taylor diagrams summarize three aspects of model performance relative to a reference: the
spatial correlation coefficient (i.e. Pearson correlation of the fields, r), the ratio of spatial
standard deviations of the two spatial fields ($\sigma_{wrf}/\sigma_{sat}$) and the root mean squared difference
(RMSD) (Taylor, 2001). Here Taylor diagrams are presented for monthly mean AOD from
WRF60, WRF12 and WRF12-remap relative to MODIS at different wavelengths (Fig. 1 d-f).
Because AOD is not normally distributed, Spearman's rank correlation coefficients ($\rho$) of the
mean monthly AOD spatial fields are also computed to reduce the impact of a few outliers and
the small sample size during cold months (Table 2). To assess the significance of $\rho$ while
accounting for multiple testing, we apply a Bonferroni correction (Simes, 1986) in which for
$m$ hypothesis tests, the null hypothesis is rejected if $p \le \dfrac{\alpha}{m}$ , where $p$ is the p-value and $\alpha$ is the
confidence level (0.05 is used here).
We further quantify the value added (or lack of thereof) of the high-resolution simulations
using the following metrics:
**(i) Brier Skill Score**
Value added is quantified using Brier Skill Scores (BSS) and is evaluated in two ways: first by
evaluating the model performance as a function of simulation resolution and then using
climatology as the reference 'forecast'. In these analyses the hourly output from the 12 km
resolution simulation is degraded (averaged) to 60 km (hereafter WRF12-remap) as follows:
the 12 km domain is resized excluding 2 grid cells at the border to exactly match the 60 km
resolution domain. For example, in the analysis of AOD each coarse grid cell thus includes 5×5
12 km resolution cells and its value is the mean of all valid 12 km grid cells inside it if at least
half of those cells contain valid AOD (i.e. no cloud cover), otherwise the whole coarse cell is
treated as missing. In all comparisons of AOD only cells with simultaneous (i.e. model and
MODIS) clear sky conditions are considered. A daily value from WRF-Chem is computed as
an instantaneous value for the hour nearest to the satellite overpass time. When the comparison
is done on a monthly basis, a monthly mean value is computed from the daily values obtained
under clear sky conditions, only if there are at least five valid observations in the month.
The primary metric used to quantify the added value of WRF12-remap versus WRF60 is the
Brier Skill Score (BSS) (Murphy and Epstein, 1989):
$$BSS = \frac{r_{F'P'}^2 - \left(r_{F'P'} - \frac{\sigma_{F'}}{\sigma_{P'}}\right)^2 - \left(\frac{\langle P'\rangle - \langle F'\rangle}{\sigma_{P'}}\right)^2 + \left(\frac{\langle P'\rangle}{\sigma_{P'}}\right)^2}{1 + \left(\frac{\langle P'\rangle}{\sigma_{P'}}\right)^2} \quad (5)$$
where $F$ is the "forecast" (i.e. the 12 km simulations mapped to 60 km, WRF12-remap); $P$ is
the "target" (i.e. for AOD this is MODIS at 60 km) and output from WRF60 are used as the
reference forecast; $F'$ the difference between 12 km estimates regridded to 60 km and MODIS;
$P'$ the difference between the 60 km simulation and the 'target' (i.e. for the AOD MODIS
observations regridded to 60 km). In the analysis of BSS relative to the long-term (15-year)
climatology of AOD from MODIS, the monthly mean climatological value of AOD is used as
the reference forecast, while WRF60 and WRF12-remap are used as the forecasts, and monthly
mean AOD from MODIS at 60 km is the target.
BSS measures by how much a test simulation (WRF12-remap) more closely (or poorly)
reproduces observations (from MODIS, MERRA-2 or other satellite products) relative to a
control (WRF60) run. For example, a BSS>0 indicates WRF12, even when regridded to 60 km,
does add value. The first term in (5) ranges from 0 to 1, is described as the potential skill, and
is the square of the spatial correlation coefficient between forecast and reference anomalies to
MODIS. It is the skill score achievable if both the conditional bias (second term) and overall
bias (third term) were zero, and for most of the variables considered herein (particularly AOD)
it contributes to a positive BSS in most calendar months (and seasons). The second term (the
conditional bias, > 0), is the square of the difference between the anomaly correlation
coefficient and the ratio of standard deviation of the anomalies and is small if for all points $F'$
is linear to $P'$. The third term is referred to as the forecast anomaly bias, and is the ratio of the
difference between the mean anomalies of WRF12-remap and the observations relative to
WRF60 and the standard deviation of WRF60 anomaly relative to observed values. The fourth
term is the degree of agreement and appears in both the numerator and denominator. It is
computed as the square of the ratio of the mean anomaly between WRF60 and observations
and the standard deviation of the anomalies.
**(ii) Pooled paired t-test**
To identify which areas in space contribute most to the AOD added-value, we compare daily
mean AOD fields from WRF-Chem at different resolutions and MODIS. We perform a pooled
paired t-test to evaluate the null hypothesis that those differences come from normal
distributions with equal means and equal but unknown variances (the test statistic has a
Student's $t$ distribution with df $= n + m - 2$, and the sample standard deviation is the pooled
standard deviation, where n and m are the two sample sizes). The test is conducted by
climatological season (e.g. winter $=$ DJF) since there are fewer than 20 valid AOD observations
in most 60 km grid cells for each calendar month (Fig. 2). Given the large number of hypothesis
tests performed (i.e. one for each 60 km grid cell), we adjust the p-values using the False
Discovery Rate (FDR) approach (Benjamini and Hochberg, 1995). In this approach, p-values
from the t-tests are ranked from low to high ($p_1, p_2, \ldots, p_m$), then the test with the highest rank, $j$,
satisfying:
$$p_j \leq \frac{j}{m} \alpha \quad \textbf{(6)}$$
is identified. Here all p-values satisfying Eq. 6 with $\alpha = 0.1$ are considered significant.
**(iii) Accuracy and Hit Rate in identification of AOD extremes**
For each month we identify grid cells in which the wavelength specific AOD exceeds the $75^{th}$
percentile value computed from all grid cells and define that as an extreme. Thus grid cells
with extreme AOD are independently determined for MODIS and WRF-Chem at different
resolutions. The spatial coherence in identification of extremes in the fields is quantified using
two metrics: the *Accuracy* and the *Hit Rate* (*HR*). The *Accuracy* indicates the overall spatial
coherence and is computed as the number of grid cells co-identified as extreme and non-
extreme between WRF-Chem and MODIS relative to the total number of cells with valid data.
The *HR* weights only correct identification of extremes in MODIS by WRF-Chem.
**3 Results**
**3.1 Model performance as a function of spatial resolution**

When WRF-Chem is applied at 60 km resolution the degree of association of the resulting spatial fields of mean monthly AOD at the three wavelengths with MODIS varies seasonally. Smallest RMSD and highest Spearman spatial correlations ($\rho$) with MODIS observations generally occur during months with highest mean AOD (i.e. during summer, Fig. 1 d-f and Fig. 3), and reach a maximum in August ($\rho = 0.60$, Table 2). However, while the patterns of relative AOD variability are well captured, the absolute magnitudes and spatial gradients of AOD during the summer are underestimated by WRF60 (Fig. 1 d-f and Fig. 3, Table S1). High spatial correlations ($\rho > 0.40$) are also observed in March, April and November (Table 2), when the ratio of spatial standard deviations is closer to 1 (Fig. 1 d-f, Table S1). Only a weak wavelength dependence is observed in the performance metrics as described on Taylor diagrams. The spatial variability is generally more negatively biased for AOD at 660 nm (Table S1), indicating that WRF60 simulations tend to produce larger diameter aerosols homogeneously distributed over the domain, whereas MODIS observations indicate more spatial variability.

The performance of WRF60 simulations relative to MODIS contrasts with analyses of WRF12 and WRF12-remap. WRF12 and WRF12-remap indicate highest spatial correlations with MODIS observations throughout the summer months ($\rho = 0.5-0.7$, Table 2), although the bias towards simulation of more coarse aerosols than are observed is consistent across the two simulations and with prior research (see details provided in (Crippa et al., 2016)). However, simulations at 12 km (WRF12) show positive $\rho$ with MODIS for all $\lambda$ in all calendar months, while mean monthly spatial fields of AOD from WRF60 show low and/or negative correlations with MODIS during May, June, September, October and December, indicating substantial differences in the degree of correspondence with MODIS AOD in the two simulations, and higher fidelity of the enhanced resolution runs (Tables 2 and S1).

Monthly mean spatial fields of AOD($\lambda$) as simulated by WRF12 or WRF12-remap exhibit positive Spearman correlation coefficients ($\rho$) with MODIS observations for all calendar months and range from ~ 0.25 for WRF12-remap (0.20 for WRF12) during winter to ~ 0.70 and 0.64, respectively during summer (Table 2). Spearman's $\rho$ are uniformly higher in WRF12-remap than WRF12 indicating a mismatch in space in the high-resolution simulation (i.e. that grid cells with high AOD are slightly displaced in the 12 km simulations possibly due to the presence of sub-grid scale aerosol plumes (Rissman et al., 2013)). Mean monthly fields of AOD (all $\lambda$) from both WRF12 and WRF12-remap exhibit lower $\rho$ with MODIS in February-April and November than the 60 km runs (Table 2). These discrepancies appear to be driven by

conditions in the south of the domain. For example, differences between WRF60/WRF12-
remap vs. MODIS during all seasons are significant according to the paired t-test over Florida
and along most of the southern coastlines (Fig. 2). This region of significant differences extends
up to ~ 40°N during summer and fall, reflecting the stronger north-south gradient in AOD from
MODIS and WRF12-remap that is not captured by WRF60 (see example for $\lambda = 550$ nm, Fig.
3). These enhancements in the latitudinal gradients from WRF12-remap are also manifest in
the physical variables (particularly specific humidity as discussed further below).
The differences in the absolute values of mean monthly AOD deriving from differences in the
resolution at which WRF-Chem was applied are of sufficient magnitude (a difference of up to
0.2 in regions with a mean AOD value of 0.4), particularly in the summer months (Fig. 4), to
raise concerns. However, detailed investigation of the simulations settings and repetition of the
60 km simulation resulted in virtually identical results indicating no fault can be found in the
analysis. Further, we note that the eastern-half of North America was also identified as a region
of high discrepancy in global ESM (Myhre et al., 2013a).
To further investigate differences in the simulation output due to spatial discretization we
computed Brier Skill Scores (BSS). In this analysis AOD for each $\lambda$ from WRF12-remap are
used as the 'forecast', output from WRF60 are used as the reference forecast and MODIS
observations at 60 km are used as the target. BSS exceed 0 during all months except for
September and October, and largest BSS ($> 0.5$) for AOD (all $\lambda$) is found during most months
between December and July (Fig. 5a-c). This indicates that running WRF-Chem at 12 km
resolution yields higher skill in simulated AOD relative to WRF60, even when the WRF12
output is remapped to 60 km. BSS do not strongly depend on $\lambda$, indicating the added value
from enhanced resolution similarly affects aerosol particles of different sizes. Inspecting the
terms defining the BSS provides information about the origin of the added value (Fig. 5a-c).
The positive BSS derives principally from the potential skill (first term in Eq. 5), which
demonstrates a reduction in bias and/or more accurate representation of the spatial gradients in
WRF12-remap. This term exhibits weak seasonality with values below 0.5 only during August
and fall months. The second and third terms are close to zero during most months, although
bigger biases are found during August-October. The substantial conditional bias during late
summer and early fall is the result of the large ratio of standard deviations ($> 1$, i.e. the spatial
variability of the anomaly relative to MODIS is larger for WRF12-remap than WRF60, Table
S1). It thus contributes to the negative BSS found in September and October, which are also
identified as outlier months in WRF12-remap from the Taylor diagram analysis (Fig. 1). Output
for these months show modest spatial correlations with AOD from MODIS and higher ratio of
standard deviations than in WRF60-MODIS comparisons (Fig. 1, Table S1). Previous work
showed that the lower model skill (in WRF12) during September and October may be partially
attributable to a dry bias in precipitation from WRF-Chem relative to observations. As a result,
simulated AOD and near-surface aerosol nitrate and sulfate concentrations are positively biased
over large parts of the domain (Crippa et al., 2016). Although the effects of the boundary
conditions appear in some variables (e.g. in Fig. 4 and Figs. S1-S3), the BSS results do not
significantly change even when those cells are removed from the analysis.
When the BSS is used to assess the skill of each model relative to MODIS AOD climatological
mean over the years 2000-2014, WRF12-remap is found to add value relative to the
climatology (i.e. BSS >0) during summer months and Nov-Jan whereas BSS for WRF60 is
positive from late Fall to early Spring (Fig. 5d). The fact that WRF-Chem does not always
outperform the climatology is expected since the model is based on time invariant emissions
and skill is assessed relative to a year selected to be representative of the AOD climatology.
Mean seasonal AOD from MODIS retrievals over the study region during 2008 lie within $\pm 0.2$
standard deviations of the climatology (Crippa et al., 2016). Interestingly, BSS for most months
(excluding September) are higher for the WRF60 simulations conducted using lateral boundary
conditions from NAM12 than GFS.
Model resolution also affects the *Accuracy* and *Hit Rate* (*HR*) for identification of areas of
extreme AOD (AOD>$75^{th}$ percentile). Highest coherence in the identification of extreme AOD
in space identified in WRF12-remap (and WRF12) relative to MODIS is found during May-
August (*HR* = 53-77%) vs. WRF60 (*HR* = 17-54%, Table 3). Conversely highest *HR* are found
for WRF60 and MODIS during winter and early spring, and indeed exceed those for WRF12
and WRF12-remap (Table 3, e.g. Feb: *HR* = 0.78 for WRF60, and 0.67 and 0.68 for WRF12
and WRF12-remap, respectively). These differences are consistent with the observation that
WRF12-remap overestimates the scales of AOD coherence and AOD magnitude during the
cold season along coastlines and over much of the domain in April (Fig. 3).
The synthesis of these analyses is thus that the higher resolution simulation increases the
overall spatial correlation, decreases overall bias in AOD close to the peak of the solar spectrum
relative to MODIS observations and therefore the higher-resolution simulations better
represent aerosol direct climate forcing. However, WRF12-remap exhibits little improvement
over WRF60 in terms of reproducing the spatial variability of AOD in the visible wavelengths
and further that WRF12-remap tends to be more strongly positively biased in terms of mean

monthly AOD outside of the summer months (Fig. 2 and Fig. 3). Also the improvement in detection of areas of extreme AOD in the higher resolution simulations (WRF12-remap) is manifest only during the warm season.

**3.2 Investigating sources of error in simulated AOD**

As documented above, WRF-Chem applied at either 60 or 12 km resolution over eastern North America exhibits some skill in reproducing observed spatial fields of AOD and the occurrence of extreme AOD values. However, marked discrepancies both in space and time are found, and at least some of them show a significant dependence on model resolution. Thus, we investigated a range of physical conditions and gas phase concentrations known to be strongly determinant of aerosol dynamics in terms of the BSS as a function of model resolution and also in terms of the mean monthly spatial patterns.

WRF12 even when remapped to 60 km provides more accurate description of key meteorological variables such as specific humidity ($Q$) within the boundary layer, *PBLH*, surface temperature and precipitation (see Fig. 6, S1, S2 and S3) when compared to MERRA-2, as indicated by the positive BSS during almost all months (Fig. 7a). Good qualitative agreement is observed for the spatial patterns and absolute magnitude of $T_{2m}$ in both WRF60 and WRF12-remap relative to MERRA-2 for all seasons (Fig. S1) leading to only modest magnitude of BSS (i.e. value added by the higher resolution simulations (Fig. 7a)). The aerosol size distribution and therefore wavelength specific AOD exhibits a strong sensitivity to $Q$ (Santarpia et al., 2005) due to the presence of hygroscopic components in atmospheric aerosols and thus the role of water uptake in determining aerosol diameter, refractivity and extinction coefficient (Zieger et al., 2013). For example, the hygroscopic growth factor, which indicates the change of aerosol diameter due to water uptake, is ~ 1.4 for pure ammonium sulfate with dry diameter of 532 nm at relative humidity of 80%, thus biases in representation atmospheric humidity may lead to big errors in simulated aerosol size and AOD (Flores et al., 2012). Our previous analyses of the 12 km resolution simulations indicated overestimation of sulfate aerosols (a highly hygroscopic aerosol component, and one which in many chemical forms exhibits strong hysteresis (Martin et al., 2004)) relative to observed near-surface $PM_{2.5}$ concentrations during all seasons except for winter (Crippa et al., 2016), leading to the hypothesis that simulated AOD and discrepancies therein may exhibit a strong dependence on $Q$. Consistent with that postulate, $Q_{PBL}$ from WRF12-remap exhibits a moist bias in cloud-free grid cells mostly during warm months, whereas WRF60 is characterized by a dry bias during all seasons (Fig. 6). Despite the positive bias, WRF12-remap better captures the seasonal

spatial patterns of $Q_{PBL}$ in MERRA-2, leading to positive BSS for this variable in all calendar
months. Thus, there is added value by higher-resolution simulations in representation of one of
the key parameters dictating aerosol particle growth and optical properties. Spatial patterns of
differences in $Q_{PBL}$ from WRF60 and WRF12-remap relative to MERRA-2 (Fig. 6) exhibit
similarities to differences in AOD (Fig. 4). WRF60 is dry-biased relative to WRF12
particularly during the summer (and fall) and underestimates $Q_{PBL}$ relative to MERRA-2 during
all seasons over the southern states and over most of continental US during summer and fall.
Conversely, WRF12-remap overestimates $Q_{PBL}$ over most of continental US during summer
and fall relative to MERRA-2.
*PBLH* is a key variable for dictating near-surface aerosol concentrations but is highly sensitive
to the physical schemes applied, and biases appear to be domain and resolution dependent.
However, this parameter is comparatively difficult to assess because differences in *PBLH* from
WRF-Chem and MERRA-2 may also originate from the way they are computed (i.e. from heat
diffusivity in MERRA-2 (Jordan et al., 2010) and from turbulent kinetic energy in WRF-Chem
(Janjić, 2002;von Engeln and Teixeira, 2013)). Nevertheless, the Mellor-Yamada-Janjich *PBL*
scheme combined with the Noah Land Surface Model applied in this work was found to
produce lower *PBL* heights (Zhang et al., 2009) than other parameterizations. Thus, the positive
bias in simulated AOD and surface $PM_{2.5}$ concentrations (reported previously in (Crippa et al.,
2016)) may be linked to the systematic underestimation of *PBLH* simulated by WRF12-remap
over continental US relative to MERRA-2 during all seasons (except winter) with greatest bias
over regions of complex topography (Fig. S2). A positive bias (of several hundred meters) in
terms of *PBLH* for WRF simulations using the MYJ parameterization was previously reported
for high-resolution simulations over complex terrain (Rissman et al., 2013), and a positive bias
in *PBLH* is also observed in the 60 km simulations presented herein (Fig. S2). This may provide
a partial explanation for the large negative bias in AOD in WRF60 during summer (Fig. 3). In
general, the BSS indicate improvement in the simulation of *PBLH* in WRF12-remap than in
WRF60 (Fig. 7a).
Consistent with the dry bias in $Q_{PBL}$ in WRF60, total accumulated precipitation is also
underestimated in WRF60, while WRF12-remap captures the absolute magnitudes and the
spatial patterns therein (Fig. S3). Analyses of hourly precipitation rates also show higher skill
for WRF12-remap than WRF60 in simulating precipitation occurrence (*HR*) relative to
MERRA-2 (Table S2). More specifically WRF12-remap correctly predicts between 40% and
70 % of precipitation events in MERRA-2 with highest skill during winter months, whereas
WRF60 output exhibits lower HR (~6% during summer and 30% during winter). This result
thus confirms our expectation of a strong sensitivity of model performance to resolution due to
the inherent scale dependence in the cumulus scheme. Use of the Grell-Freitas parameterization
in the WRF60 simulations did not lead to substantially different magnitude and/or spatial
patterns of precipitation compared to WRF60 applied with the Grell 3D scheme, and no
improvement in agreement with output from MERRA2. The findings of a negative bias in
precipitation amounts in WRF60 simulations without a corresponding overestimation of AOD
may appear counter-intuitive since aerosol concentrations (and thus AOD) are dependent on
aerosol residence times and analyses of sixteen global models from the AeroCom project
indicate wet scavenging is the dominant removal process for most aerosol species in the study
area (Hand et al., 2012;Textor et al., 2006). However, the negative precipitation bias in WRF60
simulations appears to also be linked to poor representation of surface moisture availability,
boundary layer humidity (Fig. 6), and ultimately aerosol water content (and hence AOD).
Gas phase concentrations (transformed into z-scores) from WRF12-remap show higher
agreement with satellite observations during almost all months, as indicated by the positive
BSS (Fig. 7b). However given the limited availability of valid satellite observations (especially
during months with low radiation intensity), the BSS are likely only robust for the summer
months for all species. Nevertheless, with the exception of $NH_3$ during June, BSS for all months
are above or close to zero indicating that on average, the enhanced resolution simulations do
exhibit higher skill in the simulation of the gas phase species even when remapped to 60 km
resolution. Further, the seasonal average spatial patterns of the total columnar concentrations,
expressed in terms of z-scores, also exhibit qualitative agreement with the satellite observations
(Fig. S4-S7).
**4 Concluding remarks**
This analysis is one of the first to quantify the impact of model spatial resolution on the spatio-
temporal variability and magnitude of meteorological and chemical parameters and how
representation of these variables impact AOD, and does so using simulations for a full calendar
year. Application of WRF-Chem at two different resolutions (60 km and 12 km) over eastern
North America for a representative year (2008) leads to the following conclusions:
-    Higher-resolution simulations improve the representation of key meteorological

variables such as temperature, near-surface specific humidity, boundary layer height

and the occurrence and amount of precipitation. Both spatial patterns and precipitation

occurrence are better captured by WRF12-remap, and particularly during the summer
542       months the specific humidity within the boundary-layer exhibits closer agreement with
a reanalysis product when WRF is applied at higher resolution. The dry bias in the low-
resolution WRF-Chem simulations (60 km) is consistent with previous research over
eastern North America, and is manifest in simulations with two different cumulus
parameterizations and two different data sets for the LBC (GFS and NAM12).

-  More accurate representation of spatial patterns and concentration of gaseous species
that either play a key role in particle formation and growth or are indicators of primary
aerosol emissions is also achieved by running WRF-Chem at high resolution.

-  Partly/largely due to the improved fidelity of key meteorological parameters and gas-
phase aerosol precursor species, higher resolution simulations enhance the fidelity of
AOD representation at and near to the peak in the solar spectrum relative to a coarser
run. At least some of the improvement in the accuracy with which AOD is reproduced
in the higher resolution simulations may be due to improved fidelity of specific
humidity and thus more accurate representation of hygroscopic growth of some aerosol
components. Spatial correlations of AOD from WRF12 and WRF12-remap with
observations from MODIS are higher than AOD from a simulation conducted at 60 km
during most months. WRF12 show positive spatial correlations with MODIS for all $\lambda$
in all calendar months, and particularly during summer ($\rho$ = 0.5-0.7). However, the
improvement in model performance is not uniform in space and time.

-  Output from WRF12 and WRF12-remap exhibit highest accord with MODIS
observations in capturing the frequency, magnitude and location of extreme AOD
values during summer when AOD is typically highest. During May-August WRF12-
remap has *Hit Rates* for identification of extreme AOD of 53-78%.

It is worthy of note that even the 12 km resolution WRF-Chem simulations exhibit substantial
differences in AOD relative to MODIS over eastern North America, and the agreement varies
only slightly with wavelength. This may be partially attributable to use of the modal approach
to represent the aerosol size distribution in order to enhance computational tractability. In this
application each mode has a fixed geometric standard deviation ($\sigma_g$), which can lead to biases
in simulated AOD in the visible wavelengths by up to 25% (Brock et al., 2016) (with the model
overestimating observations if the prescribed $\sigma_g$ is larger than the observed one). Setting $\sigma_g$ =
2 for the accumulation mode (the default in WRF-Chem) may lead to an overestimation of the
number of particles at the end of the accumulation mode tail, and there is evidence that a value
of $\sigma_{g,acc}$=1.40 leads to higher agreement with observations (Mann et al., 2012). Further possible
sources of the AOD biases reported herein derive from selection of the physical schemes (e.g.
planetary boundary layer (*PBL*) schemes and land-surface model (Misenis and Zhang,
2010;Zhang et al., 2009)). Further, it is worth mentioning that NEI emissions are specified
based on an average summertime weekday, so enhanced model performance might be achieved
if seasonally varying emissions were available.
Naturally, there is a need for more research regarding the sensitivity of WRF-Chem simulations
of climate relevant aerosol properties to the parameterizations used, the lateral boundary
conditions employed and the resolution at which the simulations are conducted. Further,
attribution of added-value in the simulation of AOD by enhanced spatial resolution is necessary
and will be facilitated by identifying simulation settings that minimize bias in the variables
affecting AOD. This research will be part of future investigations.

## Acknowledgments

This research was supported in part by a L'Oréal-UNESCO UK and Ireland Fellowship For
Women In Science (to PC), the Natural Environmental Research Council (NERC) through the
LICS project (ref. NE/K010794/1), grants to SCP from US NSF (grant # 1517365) and NASA
(NNX16AG31G), and a NASA Earth and Space Science Fellowship Program - Grant "14-
EARTH14F-0207" (to RCS). Further support was provided by the Lilly Endowment, Inc.,
through its support for the Indiana University Pervasive Technology Institute and the Indiana
METACyt Initiative. We gratefully acknowledge the NASA scientists responsible for
MERRA-2 and MODIS products, the developers of WRF-Chem, and Lieven Clarisse, Simon
Whitburn, and Martin Van Damme for producing and sharing the $NH_3$ retrievals. The clarity
and content of this manuscript was substantially improved by the comments of three reviewers.

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

**Figure 1. Probability density function of once daily AOD at a wavelength (λ) of 550 nm for (a) MODIS, (b) WRF60 and (c) WRF12 and WRF12-remap during the year 2008. (d-f) Taylor diagrams of mean monthly AOD at wavelengths (λ) of (d) 470, (e) 550 and (f) 660 nm as simulated by WRF-Chem at different resolutions (black diamonds=WRF60 and red dots=WRF12-remap) relative to MODIS observations. The numbers by each symbol denote the calendar month (e.g. 1=January).**

875

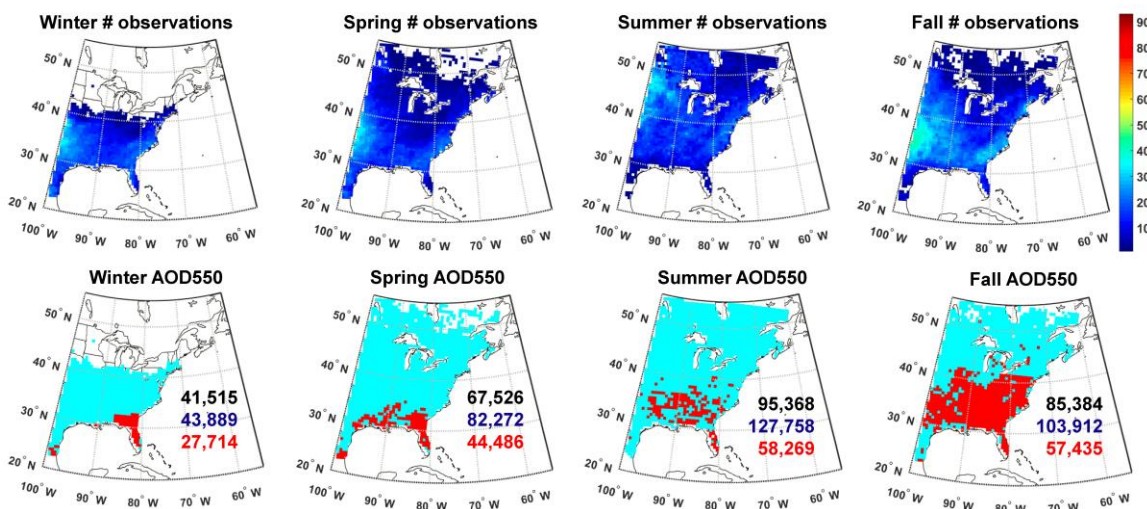

**Figure 2. First line: Number of paired AOD observations at a wavelength (λ) of 550 nm (i.e. simultaneous values as output from WRF-Chem and observed by MODIS) used to perform a t-test designed to evaluate whether the difference computed for each grid cell as WRF60-MODIS differs from that computed as WRF12-remap-MODIS on a seasonal basis (columns show Winter (DJF), Spring (MAM), Summer (JJA) and Fall (SON)). Second line: Results of the t-test. Pixels that have p-values that are significantly different at α=0.10 are indicated in red and have been corrected for multiple testing using a False Discovery Rate approach. The number of observations of cloud-free conditions summed across all days in each season and all grid cells is also reported (black=MODIS, blue=WRF60, red=WRF12-remap).**

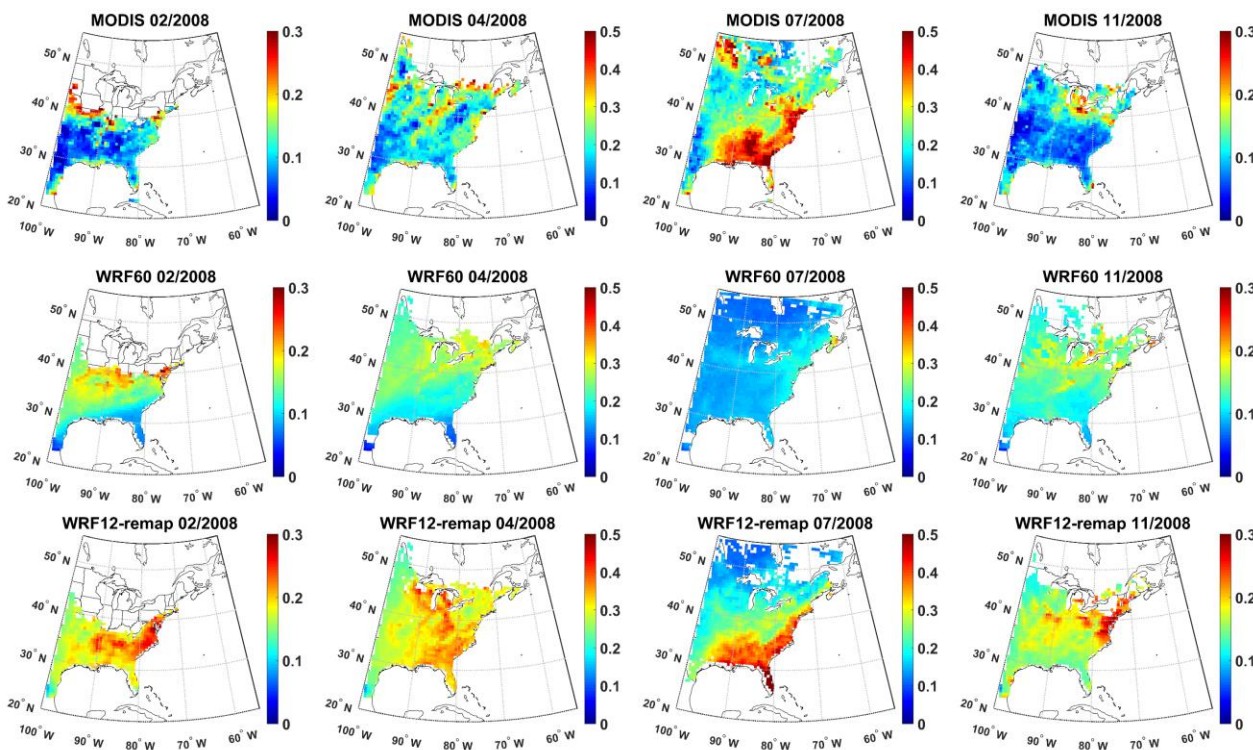

888

**Figure 3. Monthly mean AOD at a wavelength (λ) of 550 nm from MODIS (first line) and WRF-Chem at different resolutions (WRF60 and WRF12-remap, second and third line) during a representative month in each climatological season (columns). Note that a different color scale is applied for different months. For a monthly mean value for a grid cell to be shown, there must be at least 5-simultaneous daily values (for the time of the satellite overpass) available.**

895

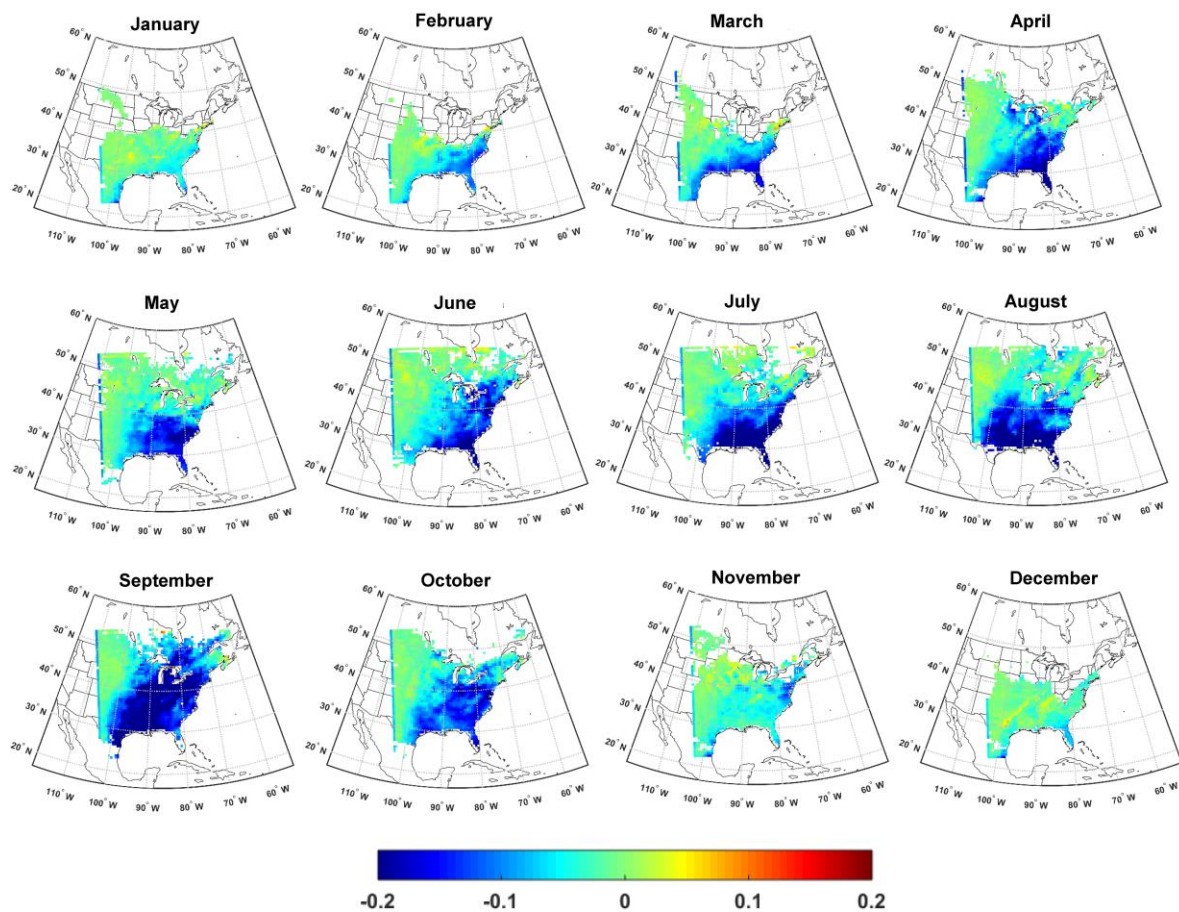

896

**Figure 4. Difference in monthly mean AOD at a wavelength (λ) of 550 nm between WRF-Chem simulations conducted at 60 km resolution (WRF60) and output from WRF-Chem simulations conducted with a resolution of 12 km but remapped to 60 km (WRF12-remap). Differences are computed as WRF60 minus WRF12-remap. Similar spatial patterns and magnitudes of differences are found for λ of 470 and 660 nm. The calendar months of 2008 are shown in the titles of each panel.**


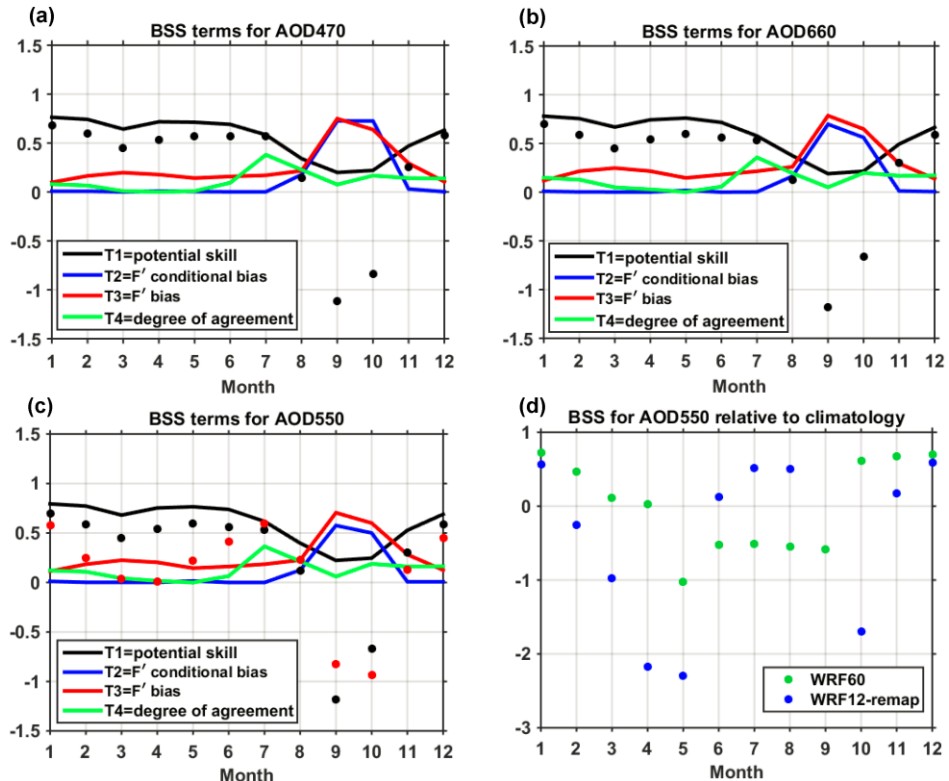


**Figure 5. (a-c) Brier Skill Scores (BSS, black dots) for monthly mean AOD by calendar**
**month (1=January) for AOD at 470, 550 and 660 nm. In this analysis of model skill**
**WRF12 output is mapped to the WRF60 grid (WRF12-remap) and BSS are computed**
**using MODIS as the target, WRF60 (driven by NAM12 meteorological boundary**
**conditions) as the reference forecast and WRF12-remap as the forecast. Also shown by**
**the color lines are the contributions of different terms to BSS. In panel c the red dots**
**indicate BSS when the reference forecast is WRF60 driven by GFS meteorological**
**boundary conditions. (d) BSS of monthly mean AOD from WRF60 (green dots) and**
**WRF12-remap (blue dots) relative to MODIS monthly mean climatology during 2000-**
**2014 (reference forecast). Monthly mean AOD from MODIS are used as the target. BSS**
**for WRF12-remap in September is -6.1.**



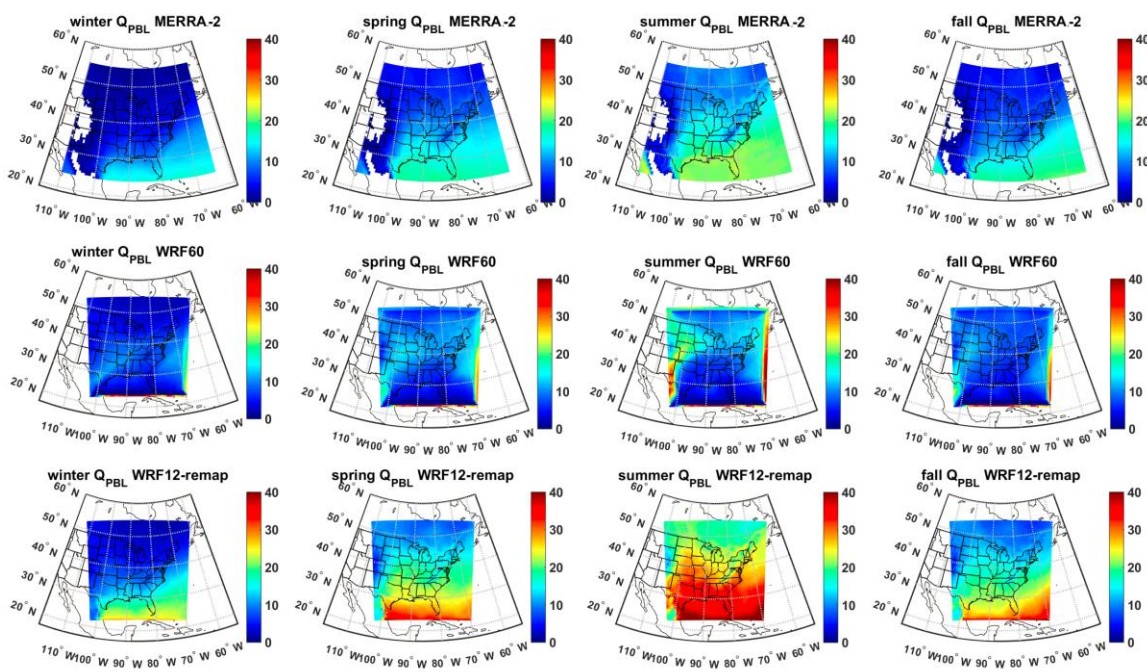


**Figure 6. Seasonal mean specific humidity [kg m$^{-2}$] integrated from the surface to 825 hPa**
**($Q_{PBL}$) from MERRA-2 (first row) assuming an average air density in the *PBL* of 1.1 kg**
**m$^{-3}$, WRF60 (second row), and WRF12-remap (third row). The data are 3-hourly and**
**show only cloud-free hours in all three data sets.**



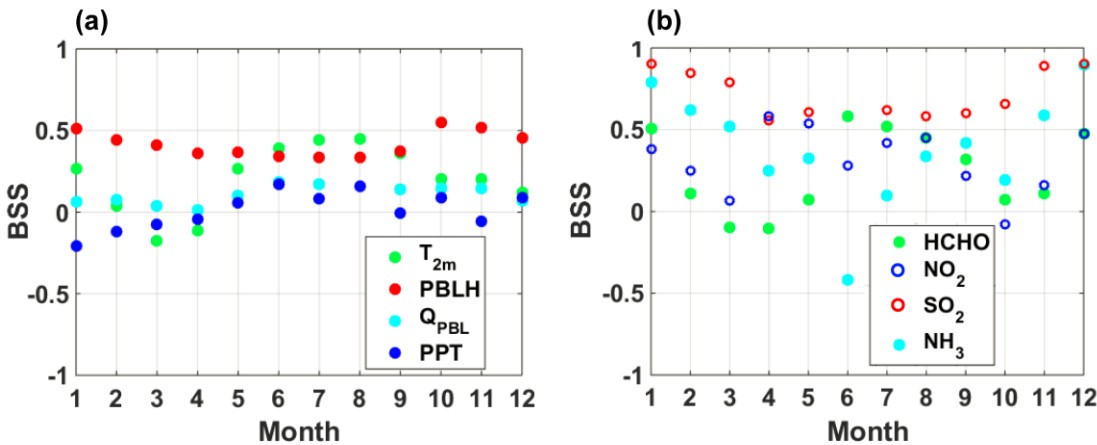


**Figure 7. Brier Skill Scores (BSS) for key (a) meteorological and (b) chemical variables. BSS are computed using hourly data of T at 2m ($T_{2m}$) and *PBLH*, 3-hourly estimates of specific humidity in the boundary layer ($Q_{PBL}$), and z-scores of monthly total precipitation (*PPT*), and of monthly mean columnar gas phase concentrations.**



**Tables**
**Table 1. Physical and chemical schemes adopted in the WRF-Chem simulations presented**
**herein.**

| Simulation settings | Values |
|---|---|
| Domain size | $300 \times 300$ $(60 \times 60)$ grid points |
| Horizontal resolution | 12 km (60 km) |
| Vertical resolution | 32 levels up to 50 hPa |
| Timestep for physics | 72 s (300 s) |
| Timestep for chemistry | 5 s |
| **Physics option** | **Adopted scheme** |
| Microphysics | WRF Single-Moment 5-class (Hong et al., 2004) |
| Longwave Radiation | Rapid Radiative Transfer Model (RRTM) (Mlawer et al., 1997) |
| Shortwave Radiation | Goddard (Fast et al., 2006) |
| Surface layer | Monin Obhukov similarity (Janjić, 2002;Janjić, 1994) |
| Land Surface | Noah Land Surface Model (Chen and Dudhia, 2001) |
| Planetary boundary layer | Mellor-Yamada-Janjich (Janjić, 1994) |
| Cumulus parameterizations | Grell 3D (Grell and Dévényi, 2002) |
| **Chemistry option** | **Adopted scheme** |
| Photolysis | Fast J (Wild et al., 2000) |
| Gas-phase chemistry | RADM2 (Stockwell et al., 1990) |
| Aerosols | MADE/SORGAM (Ackermann et al., 1998;Schell et al., 2001) |
| Anthropogenic emissions | NEI (2005) (US-EPA, 2009) |
| Biogenic emissions | Guenther, from USGS land use classification (Guenther et al., 1994;Guenther et al., 1993;Simpson et al., 1995) |



**Table 2. Spearman correlation coefficients (ρ) between AOD at wavelengths (λ) of 470, 550 and 660 nm from MODIS observations averaged over 12 or 60 km and WRF-Chem simulations conducted at 60 km (WRF60, shown in the table as -60), at 12 km (WRF12, shown in the table as -12), and from WRF-Chem simulations at 12 km but remapped to 60 km (WRF12-remap, shown in the table as -remap). Given WRF12-remap is obtained by averaging WRF12 when at least half of the 5×5 12 km resolution cells contain valid data, ρ from WRF60 and WRF12-remap may be computed on slightly different observations and sample size. The bold text denotes correlation coefficients that are significant at α=0.05 after a Bonferroni correction is applied (i.e. $p \leq \dfrac{0.05}{9 \times 12} = 4.63 \times 10^{-4}$ is significant). The yellow shading is a visual guide that shows for each month and λ the model output that has highest ρ with MODIS.**

| Month→/ Variable↓ | Jan | Feb | Mar | Apr | May | Jun | Jul | Aug | Sep | Oct | Nov | Dec |
|---|---|---|---|---|---|---|---|---|---|---|---|---|
| 470-12 | **0.238** | **0.150** | **0.137** | **0.147** | **0.377** | **0.581** | **0.610** | **0.723** | **0.352** | **0.306** | **0.259** | **0.212** |
| 470-60 | 0.156 | **0.226** | **0.438** | **0.412** | -0.219 | -0.146 | **0.379** | **0.601** | 0.087 | -0.051 | **0.500** | -0.059 |
| 470-remap | **0.295** | **0.197** | **0.250** | **0.182** | **0.516** | **0.637** | **0.675** | **0.777** | **0.368** | **0.441** | **0.315** | **0.274** |
| 550-12 | **0.223** | **0.124** | **0.142** | **0.146** | **0.349** | **0.541** | **0.580** | **0.689** | **0.275** | **0.301** | **0.280** | **0.215** |
| 550-60 | 0.179 | **0.244** | **0.429** | **0.332** | -0.288 | -0.188 | **0.324** | 0.567 | 0.073 | -0.077 | **0.491** | 0.002 |
| 550-remap | **0.297** | 0.164 | **0.261** | **0.199** | **0.493** | **0.605** | **0.651** | **0.747** | **0.286** | **0.437** | **0.352** | **0.309** |
| 660-12 | **0.217** | **0.136** | **0.165** | **0.152** | **0.324** | **0.476** | **0.540** | **0.644** | **0.183** | **0.290** | **0.292** | **0.221** |
| 660-60 | **0.191** | **0.230** | **0.437** | **0.402** | -0.305 | -0.189 | **0.389** | **0.616** | 0.099 | -0.137 | **0.536** | 0.049 |
| 660-remap | **0.356** | **0.211** | **0.289** | **0.208** | **0.480** | **0.624** | **0.669** | **0.772** | **0.371** | **0.432** | **0.393** | **0.368** |

**Table 3. Spatial coherence in the identification of extreme AOD values (i.e. areas with AOD>75[th] percentile over space for each month) between WRF-Chem at different resolutions relative to MODIS. No significant wavelength dependence is found for model skill in identifying extreme AOD so results are only shown for λ = 550 nm. The different model output is denoted by -60 for simulations at 60 km, -12 for simulations at 12 km resolution, and as –remap for simulations at 12 km but with the output remapped to 60 km. The *Accuracy* (Acc) indicates the fraction of grid cells co-identified as extremes and non-extremes between WRF-Chem and MODIS relative to the total number of cells with valid data. The *Hit Rate* (*HR*) is the probability of correct forecast and is the proportion of cells correctly identified as extremes by both WRF-Chem and MODIS. The yellow shading indicates the model resolution with highest skill in each month for AOD at 550 nm.**

| Month→/ Metric↓ | Jan | Feb | Mar | Apr | May | Jun | Jul | Aug | Sep | Oct | Nov | Dec |
|---|---|---|---|---|---|---|---|---|---|---|---|---|
| Acc-12 | 0.673 | 0.665 | 0.659 | 0.638 | 0.710 | 0.800 | 0.855 | 0.839 | 0.666 | 0.679 | 0.723 | 0.661 |
| Acc-60 | 0.707 | 0.778 | 0.735 | 0.730 | 0.600 | 0.587 | 0.658 | 0.769 | 0.661 | 0.637 | 0.729 | 0.681 |
| Acc-remap | 0.674 | 0.680 | 0.694 | 0.640 | 0.766 | 0.824 | 0.887 | 0.837 | 0.667 | 0.699 | 0.767 | 0.641 |
| HR-12 | 0.346 | 0.331 | 0.319 | 0.275 | 0.421 | 0.599 | 0.711 | 0.678 | 0.333 | 0.358 | 0.447 | 0.323 |
| HR-60 | 0.417 | 0.558 | 0.471 | 0.460 | 0.200 | 0.173 | 0.315 | 0.538 | 0.321 | 0.274 | 0.458 | 0.364 |
| HR-remap | 0.350 | 0.361 | 0.387 | 0.281 | 0.532 | 0.649 | 0.775 | 0.674 | 0.333 | 0.399 | 0.535 | 0.284 |