# Peer review of "The impact of resolution on meteorological, chemical and 1"

_Atmospheric Chemistry and Physics, 2016_

## Referee Comment (RC1) · Anonymous Referee #1 · 9 Jul 2016

Review of "Value added by high-resolution regional simulations of climate-relevant aerosol properties" by P. Crippa, R. C. Sullivan, A. Thota, S. C. Pryor

The study by Crippa et al. assesses possible improvements in high resolution simulations of aerosol by comparing aerosol optical depth (AOD) and aerosol precursor gases in two otherwise identical WRF-Chem simulations at 12 and 60 km horizontal resolution over eastern North America to MODIS for AOD and OMI/IASI for the precursor gases. The agreement of the simulations to observations in spatial patterns and extreme values are analyzed. This topic is well within the scope of Atmospheric Chemistry and Physics and the relatively long simulation period of one year could give insights whether improvements in high resolution simulations depend on season. Due

to the large differences between the 12 km and the 60 km simulation, which are not aerosol related, the very low precipitation rates in the 60 km simulation and a problem with one of the analysis methods publication can only be recommend after major revisions.

General comments:

1) Differences in meteorological variables, in particular relative humidity are identified in the paper as the main source of difference in the AOD simulation between 12 km and 60 km horizontal resolution. As the focus of the study is on improvements in the simulations of the aerosol at high resolution, the differences in meteorological variables would need to be as small as possible. Otherwise the quality of simulating meteorology is analyzed rather than aerosol. Assessing AOD and precursor gases in cloud free scenes may prove useful if the differences in meteorological variables can be minimized.

2) While the 12 km resolution simulation agrees fairly well with reanalysis data, the 60 km simulation shows large anomalies, in particular precipitation is very low. The annual mean precipitation in the studied region should be around 800 -1200 mm with a standard deviation of 180 – 260 mm (Groisman and Easterling, 1994). The precipitation of the 60 km simulation in Fig. S3 is significantly below these values in many areas. It needs to be checked if this is due to internal variability (e.g. by varying initial conditions), resolution dependent model parameters or whether one of the parameterizations used is not applicable for the resolutions used in the study.

3) In the computation of the Brier Skill Score (BSS) MODIS is used as the climatological mean and WRF60 as the current observation. This means if for example WRF60 would simulate unrealistic values, the ability of WRF12-remap is tested in this case to reproduce the unrealistic values, which is meaningless. Rather two BSS should be computed for each of the two simulations (WRF60 and WRF12-remap) where MODIS is used as the current observation and seasonal or annual mean values of MODIS are used for the climatological mean.

4) The climatological relevance of the results is not shown although the study is motivated by the uncertainty in aerosol forcing. A better accuracy for simulating the regional distribution and extreme values of AOD is important for air quality. If the same is true for effects of aerosol on radiation, clouds or precipitation is not straightforward and it would be a valuable addition if this would be assessed.

Specific comments:

P4, L73: Other studies that quantify the impact of model resolution on AOD should be discussed here e. g. Qian et al. (2010), Gustafson et al. (2011). In parallel to this study also a paper by Weigum et al. appeared on ACPD for discussion.

P4, L93: Table S1 gives relevant details of the simulations and should be moved into the main text. References for the parameterizations should be added in Table S1.

P5, L124-L130: According to Tomasi et al. (1983) alpha is often not proportional to ny-2 in the atmosphere. Furthermore, the Junge power law used in Eq. (3) is mainly interesting for historical reasons (Schuster et al., 2006) and the atmospheric aerosol size distribution is rather described by four log-normal size distributions (modes), where not all modes are present all the time in the atmosphere. But this is not particularly relevant here and the information in this paragraph should rather be that fine mode particles have smaller AOD at shorter wavelengths (e.g. 440 nm) than at longer wavelengths (e.g. 865 nm) whereas for coarse mode particles AOD is similar at shorter and longer wavelengths. This is reflected in the Angstrom parameter and the Angstrom parameter can therefore be used as a proxy for the fine mode fraction or fine mode radius (depending on the definition, see Schuster al. 2006).

P6, L144: For which year are the anthropogenic aerosol emissions, 2005, 2008, 2009? If not 2008, why is 2008 simulated and not the year corresponding to the aerosol emissions?

P6, L152: Are the cells at the outer boarder of the domain excluded from the analysis?

In some Figures e. g. Fig. 4, Fig. 6, Figs. S1-S3 one can clearly see the effects of the boundary conditions.

P6, L157-160: This is not clear. Is a single, instantaneous value used at the time of the satellite overpass or are several time steps averaged around the time of the satellite overpass. If the latter: how many time steps, in which time period?

P7, L172-175: Given the uncertainty of MODIS observations is there a minimum value for AOD used for the analysis? BSS incorporates the uncertainty in the observations but what about the other methods used?

P8, L198: Different definitions are used in the literature for planetary boundary height (PBLH), which can result in large differences in PBLH (e. g. von Engeln and Teixeira, 2013). Are the definitions for PBLH in MERRA-2 and WRF-Chem the same?

P11, L314: No explanation is given why BSS is so small in September and October (Fig. 5). Also in Fig. 1 d)-f) the standard deviation of September and October of WRF12-remap is much larger than for the other months. What is the reason for this?

P13, L370-375: How does AOD without AOD from aerosol water compare between WRF12-remap and WRF60?

P13, L377: What is the reason of the dry bias (also over the ocean) in WRF60?

P24, Fig. 3: Why are monthly values shown and not seasonal values as in the other Figures?

-, -: It should be mentioned clearly in the text that the analysis is conducted only over land and discussed why this is done.

Technical corrections:

P1, L1: The relevance for climate of the results is unclear so the title should rather be "Value-added by high-resolution regional simulations of aerosol properties"

P3, L51-52: References for the forcing estimates are missing.

P4, L76: Diaconescu and Laprise (2013) note that "the main added value of an RCM is provided by its small scales and its skill to simulate extreme events, particularly for precipitation." As this is relevant for the current study it could be mentioned in the text.

P5, L118: Eq. (2) can be derived from Eq. (1) by integration over the atmospheric optical path. It would be clearer if lambda_1 and lambda_2 are also used in Eq. (1) instead of lambda and lambda=1 micrometer.

P5, L121: Define Dp.

P6, L128: Which geometric standard deviation is used for the coarse mode?

P6, L132-160: The model description should be expanded, in particular the part relevant for the aerosol simulation.

P6, L139: The total number of layers should be mentioned here as well.

P7, L162-L183: Give more details about the satellite products used e.g. resolution, coverage etc.

P7, L173-174: Give the right uncertainty values i.e. (+-0.05 +15%) and (+-0.05+15-20%).

P7, L184-L187: Reformulate to explain better how the regridding is done.

P7, L190: Standard scores could be shortly explained.

P8, L206-207: The root mean square difference is not shown in Fig. 1 a)-c).

P9, L225-239: This could be explained better. In Murphy and Epstein it is noted that the first term would be the skill if the second and third term were small. The second term is small if for all points F' is linear to P' (conditional bias). The third term gives the overall/mean bias. The fourth term is a correction and should be small.

P23, Fig. 2: It would be useful to add the number of cloud-free data points for each

season and each of the three datasets (WRF12-remap, WRF60, MODIS).

References:

von Engeln and Teixeira, 2013, J. Climate, doi:10.1175/JCLI-D-12-00385.1

Groisman and Easterling, 1994, J. Climate, doi: 10.1175/1520-0442(1994)007<0184:VATOTP>2.0.CO;2

Gustafson et al. , 2011, J. Geophys. Res., doi: 10.1029/2010JD015480

Qian et al. , 2010, Atmos. Chem. Phys., doi: 10.5194/acp-10-6917-2010

Schuster et al., 2006, J. Geophys. Res., doi: 10.1029/2005JD006328

Weigum et al., 2016, Atmos. Chem. Phys. Discuss., doi: 10.5194/acp-2016-360
* * *

---

## Referee Comment (RC2) · Anonymous Referee #2 · 28 Jul 2016

**Summary**

The present work investigates how an increase in spatial (horizontal) resolution from 60 km to 12 km improves the ability of a mesoscale simulation with the WRF-Chem model to reproduce satellite observations of aerosol optical depth (AOD) and column concentrations of chemical species, and of key meteorological variables (relative to a reanalysis product). The WRF-Chem model is run for the year 2008 on a domain that covers the eastern USA, a region with strong emissions of natural and anthropogenic aerosol precursors.

The motivation of the work is established in the introduction. Simulations, observations,

and reanalysis data are subsequently introduced with select relevant details, such as uncertainties in the satellite observations. The authors then provide an overview of the statistical methods used to evaluate model performace relative to observations and reanalysis in a succint but effective way, which can serve as an introduction to novices to the subject. In the main part of the work the results are presented and analyzed, and the authors carefully quantify and discuss the improvements from an increase in resolution. It is found that a higher simulation resolution improves model fidelity in reproducing observed AOD as well as the ability of the model to identify extreme AOD values. The analysis reveals that the improved performance of the model at higher resolution can in part be attributed to improved agreement in meteorological quantities, in particular boundary layer specific humidity, which contributes to aerosol growth. The model is also shown to better reproduce satellite observations of chemical species at higher resolution. The authors do not fail to identify instances where a higher resolution does not result in an improvement: While model skill (measured by the Brier skill score) in reproducing AOD improves in seven out of twelve months, the model shows improvements in detecting extreme AOD values only in the warm season. In all this, statistical methods are used effectively to quantify model performance.

The work addresses an interesting question of general importance in atmospheric modeling - what improvements can be expected from an increase in model resolution? The authors answer this question in exemplary fashion for aerosol properties, chemistry, and meteorology in a mesoscale model. The key insight is that increased spatial (horizontal) resolution clearly improves model performance but is not a panacea. Model aspects other than resolution need attention as well to improve model fidelity.

The manuscript is thorough, clear, compelling, very well written, and presents the results with good figures and tables. I recommend publication after attending to the following detailed comments.

**Detailed comments**

Line 51-52: Please give references for the possible range of values for the direct and indirect aerosol effects.

Section 2.1: Aerosol size distributions are generally log-normal, not power law functions. The upper tail of a log-normal distribution does, however, behave like a power law distribution. The discussion here is therefore not incorrect, however, it should be modified to assume a log-normal distribution (also because the aerosol scheme used in the WRF-Chem simulations assumes log-normal distributions).

Line 417-418: "Further, the seasonal average spatial patterns of the total columnar concentrations, expressed in terms of z-scores, also exhibit high qualitative agreement with the satellite observations (Fig. S4-S7)."

It is a stretch to write "high qualitative agreement" here. Comparing the OMI, WRF60, and WRF12-remap panels in Fig. S4-S7, my impression is that omitting "high" and leaving "qualitative agreement" is a more accurate assessment.

Table 1: A factor of 9 is placed in the denominator of the Bonferri correction, but the data at the different wavelenghts, resolutions, and remappings would not seem to be truly independent significance tests. For example, WRF12 and WRF12-remap would seem to be very dependent. Please carefully consider (and justify) whether the factor of 9 should be used or rather omitted.

Figure S5: OMI, WRF60, and WRF12-remap panels (or panel titles) are shuffled relative to the panel order in Figures S4, S6, and S7.

---

## Author Comment (AC1) · 19 Aug 2016

**Response to review comments on acp-2016-453 from reviewer 1**

**The original comments are provided in black, our response is given below each comment in red.**

**Thank you for the careful reading of our manuscript and your review.**

The study by Crippa et al. assesses possible improvements in high resolution simulations of aerosol by comparing aerosol optical depth (AOD) and aerosol precursor gases in two otherwise identical WRF-Chem simulations at 12 and 60 km horizontal resolution over eastern North America to MODIS for AOD and OMl/IASI for the precursor gases. The agreement of the simulations to observations in spatial patterns and extreme values are analyzed. This topic is well within the scope of Atmospheric Chemistry and Physics and the relatively long simulation period of one year could give insights whether improvements in high resolution simulations depend on season. Due to the large differences between the 12 km and the 60 km simulation, which are not aerosol related, the very low precipitation rates in the 60 km simulation and a problem with one of the analysis methods publication can only be recommend after major revisions.

**Thank you for your positive assessment. We have addressed the general and specific comments below and modified the manuscript accordingly.**

**General comments:**

1) Differences in meteorological variables, in particular relative humidity are identified in the paper as the main source of difference in the AOD simulation between 12 km and 60 km horizontal resolution. As the focus of the study is on improvements in the simulations of the aerosol at high resolution, the differences in meteorological variables would need to be as small as possible. Otherwise the quality of simulating meteorology is analysed rather than aerosol. Assessing AOD and precursor gases in cloud free scenes may prove useful if the differences in meteorological variables can be minimized.

**Thanks for the comment. We agree that meteorological variables play a key role in dictating aerosol and gas properties, thus an accurate simulation of those variables naturally will help in reproducing satellite observations of aerosol properties. In response to your comments we expanded the literature review on the added value at line 74 (please refer to our specific answer below). E.g. We are aware of the work of Weigum et al (in review for ACP, and indeed we cite that work in our paper) and think that their attempts to decompose the performance are interesting. Our focus is slightly different - we aim to quantify the value added only by enhanced resolution to the meteorology, gas phase concentrations and the aerosol properties, we are not seeking to evaluate (per se) changes in physical parameterizations. Thus it is essential that we do not change parameterizations between the runs, and we have elected to use the parameterizations that prior research has demonstrated is appropriate at high resolutions (e.g. using a convective parameterization intended for near 'gray zone' resolution simulations) (Grell and Dévényi, 2002;Nasrollahi et al., 2012;Crippa et al., 2016). Our analysis indicates that the improved skill of the high-resolution simulations in reproducing AOD is driven by the skill in reproducing BOTH the meteorological and chemical fields via better representation of fine scale aerosol dynamics.**

**Naturally, there is a lot more work to be done. We are currently conducting a broader analysis to investigate meteorological, chemical and aerosol properties'**

**sensitivity to different parameterizations at different resolutions that will complement results presented in this work, but it is beyond the scope of this paper.**

2) While the 12 km resolution simulation agrees fairly well with reanalysis data, the 60 km simulation shows large anomalies, in particular precipitation is very low. The annual mean precipitation in the studied region should be around 800 -1200 mm with a standard deviation of 180 - 260 mm (Groisman and Easterling, 1994). The precipitation of the 60 km simulation in Fig. 53 is significantly below these values in many areas. It needs to be checked if this is due to internal variability (e.g. by varying initial conditions), resolution dependent model parameters or whether one of the parameterizations used is not applicable for the resolutions used in the study.

**We agree that the 12 km simulations perform better than WRF60 for most of the meteorological, chemical and aerosol components and that a big bias is present in the precipitation fields simulated by WRF60. The choice of the adopted parameterizations is based on our previous work and evaluation (Crippa et al., 2016), which showed good skill of WRF12 in reproducing aerosol optical properties. Therefore the current study aims to verify if the increased resolution (i.e. from 60 km to 12 km) played a role in a more accurate description of simulated properties relative to observations.**

**The reviewer is quite correct in identifying precipitation bias as a key challenge in regional modelling (both physical and coupled with chemistry). For example, the NARCCAP simulations with WRF at 50-km were also dry biased in the study domain. Although there have been a number of studies that have sought to evaluate different cumulus schemes over different regions at different resolutions, to our knowledge no conclusion (definitive recommendation) has been made regarding the dependence of model's skill on resolution and cumulus parameterization (Arakawa, 2004;Jankov et al., 2005;Nasrollahi et al., 2012). A strong sensitivity on the adopted cumulus scheme was found in (Li et al., 2014), where the Grell 3 scheme is responsible for a wet bias in the Southeast US (mostly in summer). In that study the model was run at 15 km resolution which the authors identified as the minimum resolution to be able to resolve the rainfall system with a 60-km spatial scale typical of the region. Further, the Grell 3D scheme was successfully applied at resolutions in the range of 1-36 km (e.g. (Grell and Dévényi, 2002;Lowrey and Yang, 2008;Nasrollahi et al., 2012;Sun et al., 2014;Zhang et al., 2016)), although further research is needed to identify the optimal cumulus scheme over North America at coarser resolution, which is part of our ongoing work.**

**Nevertheless, the reviewer's comments have prompted us to include a great deal more discussion of the possible sources of these discrepancies, linking to the adopted schemes and to the potential bias based on other sensitivity studies, and to the number of simulated cloud free grid cells at different resolutions. It would be very interesting to see the sensitivity of the model to varying initial conditions (e.g. using a different reanalysis product for initial conditions), but as the reviewer notes we are one of the first groups to attempt such long (computationally expensive) simulations and are not currently able to rerun the simulations with variable initial conditions.**

3) In the computation of the Brier Skill Score (BSS) MODIS is used as the climatological mean and WRF60 as the current observation. This means if for example WRF60 would simulate unrealistic values, the ability of WRF12-remap is tested in this case to reproduce the unrealistic values, which is meaningless. Rather two BSS should be computed for each of the two simulations (WRF60 and WRF12-remap) where MODIS is used as the current observation and seasonal or annual mean values of MODIS are used for the climatological mean.

**An alternative definition for the BSS to the one reported in the manuscript (equation 4) is the following:**

$$BSS = 1 - \frac{BS_F}{BS_{ref}}$$

**where BS$_F$ and BS$_{ref}$ are the Brier Scores of the forecast (i.e. in our case WRF12-remap vs MODIS) and the reference (WRF60 vs MODIS).**
**The Brier Score can be computed as:**

$$BS = \frac{1}{N}\sum_{i=1}^{N}\left(p_i - o_i\right)^2$$

**where $N$ is the sample size, $o_i$ are the observations (i.e. MODIS) and $p_i$ are the simulated values (i.e. either WRF60 or WRF12-remap). The whole derivation from the BSS reported here to the one in the manuscript can be found in (Murphy and Epstein, 1989). Given the BSS is based on the relative comparison of different simulations to the same reference (i.e. MODIS, as stated in the discussion manuscript from line 221), we believe it is an appropriate metric to quantify the improvement of using high versus coarse resolution.**
**For clarity, we rephrased at line 260 as follows:**
**"BSS measure how much a test simulation (i.e. WRF12-remap) more closely (or poorly) reproduces observations (from MODIS, MERRA-2 or other satellite products) relative to a control (WRF60) run."**

4) The climatological relevance of the results is not shown although the study is motivated by the uncertainty in aerosol forcing. A better accuracy for simulating the regional distribution and extreme values of AOD is important for air quality. If the same is true for effects of aerosol on radiation, clouds or precipitation is not straightforward and it would be a valuable addition if this would be assessed.
**The reviewer is quite correct, but we are clear (in the title and elsewhere) that we aim to quantify the value added by high resolution in simulating "climate-relevant aerosol properties" and not the added value in describing climate forcing due to aerosols. Therefore we decided to keep the original title (in response to the specific comment below) and devote further studies to investigate the possible reduction in aerosol climate forcing uncertainty due to the enhanced resolution.**

**Specific comments:**
P4, L73: Other studies that quantify the impact of model resolution on AOD should be discussed here e. g. Qian et al. (2010), Gustafson et al. (2011). In parallel to this study also a paper by Weigum et al. appeared on ACPD for discussion.
**Thanks for the useful references. We added the following discussion on them at line 74:**

**"There is empirical evidence to suggest strong resolution dependence in simulated aerosol particle properties. For example, WRF-Chem simulations with spatial resolution enhanced from 75 km to 3 km exhibited higher correlations and lower bias relative to observations of aerosol optical properties over Mexico likely due to more accurate description of emissions, meteorology and of the physicochemical processes that convert trace gases to particles (Gustafson et al., 2011;Qian et al., 2010). This improvement in the simulation of aerosol optical properties implies, a reduction of the uncertainty in associated aerosol radiative forcing (Gustafson et al., 2011). Further,**

**WRF-Chem run over the United Kingdom and Northern France at multiple resolutions in the range of 40-160 km, underestimated AOD by 10-16% and overestimated CCN by 18-36% relative to a high resolution run at 10 km, partly as a result of scale dependence of the gas-phase chemistry and differences in the aerosol uptake of water (Weigum et al., 2016)."**

P4, L93: Table S1 gives relevant details of the simulations and should be moved into the main text. References for the parameterizations should be added in Table S 1.
**Thanks, done.**

P5, L 124-L 130: According to Tomasi et al. (1983) alpha is often not proportional to ny-2 in the atmosphere. Furthermore, the Junge power law used in Eq. (3) is mainly interesting for historical reasons (Schuster et al., 2006) and the atmospheric aerosol size distribution is rather described by four log-normal size distributions (modes), where not all modes are present all the time in the atmosphere. But this is not particularly relevant here and the information in this paragraph should rather be that fine mode particles have smaller AOD at shorter wavelengths (e.g. 440 nm) than at longer wavelengths (e.g. 865 nm) whereas for coarse mode particles AOD is similar at shorter and longer wavelengths. This is reflected in the Angstrom parameter and the Angstrom parameter can therefore be used as a proxy for the fine mode fraction or fine mode radius (depending on the definition, see Schuster al. 2006).
**Thanks for this comment. We rephrased as follows and added the citation of Schuster et al., 2006.**

**The relationship between the aerosol size distribution and spectral dependence of AOD is described by a power law function:**

$$\beta(\lambda_1) = \beta(\lambda_2) \times \frac{\lambda_1}{\lambda_2}^{-\alpha} \quad \textbf{(1)}$$

**where $\beta$ is the particle extinction coefficient at a specific wavelength $\lambda$ and $\alpha$ is the Ångström exponent (Ångström, 1964) which describes the wavelength dependence of AOD (and is inversely proportional to the average aerosol diameter):**

$$\alpha = \frac{\ln \frac{AOD(\lambda_1)}{AOD(\lambda_2)}}{\frac{\lambda_2}{\lambda_1}} \quad \textbf{(2)}$$

**The aerosol volume distribution (and thus also its size distribution) usually conforms to a multi-lognormal function with $n$ modes:**

$$\frac{dV(r)}{d\ln r} = \sum_{i=1}^{n} \frac{C_i}{\sqrt{2\pi}\sigma_i} \exp\left[ \frac{-(\ln r - \ln R_i)^2}{2\sigma_i^2} \right] \quad \textbf{(3)}$$

**where $C_i$ is the particle volume concentration in the mode $i$, $R_i$ is the geometric mean radius and $\sigma_i$ is the geometric standard deviation, thus we have:**

$$AOD(\lambda) = \int \frac{3\beta(m,r,\lambda)}{4r} \frac{dV(r)}{d\ln r} d\ln r dZ \quad \textbf{\textcolor{red}{(4)}}$$

**As indicated in (Schuster, 2006), "the spectral variability of extinction diminishes for particles larger than the incident wavelength", thus fine mode particles contribute more to AOD in the visible (λ~0.5 μm) than at longer wavelengths, whereas coarse mode particles provide a similar AOD both at short and long wavelengths. This is reflected in the Ångström parameter which can be thus used as a proxy for the fine mode fraction or fine mode radius (Schuster, 2006).**

P6, L 144: For which year are the anthropogenic aerosol emissions, 2005, 2008, 2009?
If not 2008, why is 2008 simulated and not the year corresponding to the aerosol emissions?
**Anthropogenic emissions are for the year 2005 since they are the closest in time to the year 2008. We are simulating the year 2008 for its climate representativeness, as assessed by other studies based on multiple sources of measurements over the area (e.g. (Crippa et al., 2016)) and for comparison with them.**
**We added the following comment from line 156:**
**"Physical and chemical parameterizations were chosen to match previous work using WRF-Chem at 12 km on the same region which showed good performance relative to observations and the year 2008 was selected because representative of average climate and aerosol conditions during 2000 - 2014 (Crippa et al., 2016)."**

P6, L 152: Are the cells at the outer boarder of the domain excluded from the analysis?
In some Figures e.g. Fig. 4, Fig. 6, Figs. S1-S3 one can clearly see the effects of the boundary conditions.
**Thanks for pointing this out. In the original manuscript, the outer cells of the domain were not excluded from the analyses. However we checked that removing either 3 or 5 cells from each side of the domain (i.e. ~180-300 km), does not significantly affect the BSS results (i.e. if present, changes in BSS occur after the fourth decimal digit). Therefore, we decided to keep the original analysis for a more clear comparison.**

P6, L 157-160: This is not clear. Is a single, instantaneous value used at the time of the satellite overpass or are several time steps averaged around the time of the satellite overpass. If the latter: how many time steps, in which time period?
**Thanks for pointing this out. We clarified at line 184 that daily values from WRF-Chem are for the hour nearest to the overpass time and that a monthly mean is computed from the daily values at the overpass time as follows:**

**"A daily value from WRF-Chem is computed as an instantaneous value for the hour nearest to the satellite overpass time. When the comparison is done on a monthly basis, a monthly mean value is computed from the daily values obtained under clear sky conditions, only if there are at least five valid observations in the month."**

P7, L 172-175: Given the uncertainty of MODIS observations is there a minimum value for AOD used for the analysis? BSS incorporates the uncertainty in the observations but what about the other methods used?
**The minimum value of AOD retrievals is -0.1, which are considered valid for near zero AOD conditions within the retrieval uncertainty; low AOD retrievals are physically representative of low aerosol concentrations (and thus removing them would bias the**

**analysis), and although low AOD may be degraded due to errors in land surface assumptions, we do not implement additional quality assurance constraints beyond those already implemented in the MODIS Level-2, Collection 6 product in order to increase the number of valid retrievals used in our analyses (Levy et al., 2013).**

**Random errors in the MODIS retrievals should not greatly impact the analyses, as any errors should decrease 'skill' equally in both WRF60 and WRF12-remap. Similarly, any systematic error in the MODIS product (e.g. due to assumptions about underlying land surface and/or predominant aerosol type (Levy et al., 2007)), should equally impact both WRF60 and WRF12-remap. As we have no a priori expectation that the different resolution simulations would have biases that coincide with that of the MODIS product, and the analysis methods used generally compare relative change in 'skill' between the different resolutions, we do not expect uncertainty in the MODIS product to significantly impact our finding.**

PB, L 198: Different definitions are used in the literature for planetary boundary height (PBLH), which can result in large differences in PBLH (e. g. von Engeln and Teixeira, 2013). Are the definitions for PBLH in MERRA-2 and WRF-Chem the same?

**MERRA PBLH is diagnosed as the level at which the heat diffusivity drops below a value of 1 $m^2$ $s^{-1}$ (Jordan et al., 2010). The Mellor-Yamada-Janjich PBL scheme adopted here predicts the turbulent kinetic energy (TKE) at every model level and has a 2.5-order turbulent closure (Janjić, 2002). The PBLH is defined as the lowest model level where the turbulence approaches its prescribed lower bound (i.e. TKE ~ 0.2 $m^2s^{-2}$). Therefore some differences are present in the way PBLH is computed between MERRA-2 and WRF-Chem which may impact our results (von Engeln and Teixeira, 2013).**

**We have now rephrased from line 433 as follows:**
**"PBLH is a key variable for dictating near-surface aerosol concentrations but is highly sensitive to the physical schemes applied, and biases appear to be domain and resolution dependent. However, differences in PBL heights between WRF-Chem and MERRA-2 may also originate from the way they are computed (i.e. from heat diffusivity in MERRA-2 (Jordan et al., 2010) and from turbulent kinetic energy in WRF-Chem (Janjić, 2002)) (von Engeln and Teixeira, 2013). The Mellor-Yamada-Janjich PBL scheme combined with the Noah Land Surface Model applied in this work was found to produce lower PBL heights (Zhang et al., 2009) than other parameterizations."**

**As indicated in previous studies, "over much of the United States and portions of the subtropical oceans, the MERRA PBL depths are within 25% of the estimates derived from CALIPSO…" although "over the arid and semiarid complex terrain of the Southwestern United States and the Rocky Mountain region, the CALIPSO retrievals estimate a relatively shallow PBL depth compared to reanalysis" (McGrath-Spangler and Denning, 2012).**

P11, L314: No explanation is given why BSS is so small in September and October (Fig. 5). Also in Fig. 1 d)-f) the standard deviation of September and October of WRF12-remap is much larger than for the other months. What is the reason for this?

**We have now added at line 370 the following explanation for the lower model performance in September and October and referred to our previous work in which we analyzed this aspect in more detail:**

**"Previous work with analogous WRF-Chem settings showed that the lower model skill during September and October can be partially attributable to a dry bias in precipitation from WRF-Chem relative to observations. As a result, a positive bias in simulated AOD and aerosol nitrate and sulfate concentrations is present over large regions of the domain (Crippa et al., 2016)."**

P13, L370-375: How does AOD without AOD from aerosol water compare between WRF12-remap and WRF60?

**This is a very interesting point. Unfortunately, we did not save aerosol water in the Aitken and accumulation mode in our output variables, but this will be certainly considered for future work.**

P13, L377: What is the reason of the dry bias (also over the ocean) in WRF60?

**As indicated by the new Figure 2, WRF60 simulates a higher number of cloud free grid cells than MODIS in all seasons and approximately twice the number of cloud free pixels of WRF12-remap, a factor that will be strongly associated with the detected dry bias. Although a dry bias is present in WRF60, we did not change parameterizations between the runs to be able to attribute differences in skills only to the enhanced resolution (please refer to our answers above).**

P24, Fig. 3: Why are monthly values shown and not seasonal values as in the other Figures?

-, -: It should be mentioned clearly in the text that the analysis is conducted only over land and discussed why this is done.

**Given this work seeks to investigate model's skill in describing MODIS AOD and given the high temporal frequency of the WRF-Chem output, all analyses (i.e. BSS, Taylor diagrams, extremes) are conducted on a monthly basis, thus also Figure 3 and 4 report differences in AOD spatial patterns and magnitude on a monthly basis. The figures on the meteorological and chemical variables in the supplementary materials are reported on a seasonal basis to allow the reader to better understand inter-seasonal changes in the spatial patterns looking at an aggregate information (reporting monthly data for the three datasets would have required a figure of 36 panels for each variable analyzed).**

**The analysis on AOD is conducted only over land since we are comparing relative to the MODIS Collection 6 dark-target land aerosol product. Retrievals of AOD over land and over ocean invoke different assumptions about surface and aerosol properties, and are thus retrieved with different uncertainty (Levy et al., 2013). Including the ocean product would have thus caused inconsistencies in the model skill assessment. We added the following at line 192:**

**"To provide a consistent assessment of model skill, the evaluation of AOD is conducted only on land areas since the MODIS dark-target ocean aerosol product is based on a retrieval algorithm different from the one over land (Levy et al., 2013)."**

**Technical corrections:**

P1, L 1: The relevance for climate of the results is unclear so the title should rather be "Value-added by high-resolution regional simulations of aerosol properties"
**We believe our title is appropriate – and thus prefer to keep it as is.**

P3, L51-52: References for the forcing estimates are missing.
**Thanks for noting this. We have now added the following reference:**
**Stocker, T. F. a. Q., D. and Plattner, G.-K. and Alexander, L.V. and Allen, S.K. and Bindoff, N.L. and Bréon, F.-M. and Church, J.A. and Cubasch, U. and Emori, S. and Forster, P. and Friedlingstein, P. and Gillett, N. and Gregory, J.M. and Hartmann, D.L. and Jansen, E. and Kirtman, B. and Knutti, R. and Krishna Kumar, K. and Lemke, P. and Marotzke, J. and Masson-Delmotte, V. and Meehl, G.A. and Mokhov, I.I. and Piao, S. and Ramaswamy, V. and Randall, D. and Rhein, M. and Rojas, M. and Sabine, C. and Shindell, D. and Talley, L.D. and Vaughan, D.G. and Xie, S.-P. (2013), Summary for Policymakers, in *Climate Change 2013: The Physical Science Basis. Contribution of Working Group I to the Fifth Assessment Report of the Intergovernmental Panel on Climate Change*, edited, pp. 33–115, Cambridge University Press, Cambridge, United Kingdom and New York, NY, USA.**

P4, L76: Diaconescu and Laprise (2013) note that "the main added value of an RCM is provided by its small scales and its skill to simulate extreme events, particularly for precipitation:' As this is relevant for the current study it could be mentioned in the text.
**We agree. We added the quote at line 94.**
**Further, "the main added value of a regional climate model is provided by its small scales and its skill to simulate extreme events, particularly for precipitation" (Diaconescu, 2013).**

P5, L 118: Eq. (2) can be derived from Eq. (1) by integration over the atmospheric optical path. It would be clearer if lambda_ 1 and lambda_2 are also used in Eq. (1) instead of lambda and lambda=1 micrometer.
**We agree. We modified the equation as follows:**

**"The relationship between the aerosol size distribution and spectral dependence of AOD is described by a power law function:**

$$\beta(\lambda_1) = \beta(\lambda_2) \times \frac{\lambda_1}{\lambda_2}^{-\alpha} \quad \text{(5)}$$

**where $\beta$ is the particle extinction coefficient at a specific wavelength $\lambda$ and $\alpha$ is the Ångström exponent (Ångström, 1964) which describes the wavelength dependence of AOD (and is inversely proportional to the average aerosol diameter)"**

P5, L 121: Define Dp.
**We removed Dp since it is not present in the new equations (please refer to our answer above).**

P6, L 128: Which geometric standard deviation is used for the coarse mode?

**The geometric standard deviation for the coarse mode is 2.5. We have now added this information in the manuscript.**

P6, L 132-160: The model description should be expanded, in particular the part relevant for the aerosol simulation.

**We have now expanded the section describing simulations settings by adding the following from line 154:**

**"Simulation settings are identical for the two runs except for the time-step used for the physics (Table 1). Physical and chemical parameterizations were chosen to match previous work using WRF-Chem at 12 km on the same region which showed good performance relative to observations and the year 2008 was selected because representative of average climate and aerosol conditions during 2000 - 2014 (Crippa et al., 2016). More specifically the simulations adopted the RADM2 chemical mechanism (Stockwell et al., 1990) and a modal representation of the aerosol size distribution (MADE/SORGAM, (Ackermann et al., 1998;Schell et al., 2001)) with three lognormal modes and fixed geometric standard deviations (i.e. 1.7, 2 and 2.5 for Aitken, accumulation and coarse mode, respectively (Ackermann et al., 1998;Grell et al., 2005)). Aerosol direct feedback was turned on and coupled to the Goddard shortwave scheme (Fast et al., 2006). A telescoping vertical grid with 32 model layers from the surface to 50 hPa and 10 layers up to 800 hPa was selected."**

P6, L 139: The total number of layers should be mentioned here as well.
**Added.**

P7, L162-L183: Give more details about the satellite products used e.g. resolution, coverage etc.
**We have explicitly stated the resolution of the satellite products in the discussion paper (lines 168-172), and have added a sentence regarding the temporal coverage of the satellite products. We have also already included details regarding overpass times, measurement uncertainty, and post-processing (e.g. cloud screening). We believe we have provided the information pertinent to our analyses, and as other papers have been dedicated to describing these products in further detail, we refer the readers to the product specific papers (e.g. reference given in section 2.3).**

**We have amended the text to include the spatial coverage of the satellite products:**

**"The MODIS algorithm removes cloud-contaminated pixels prior to spatial averaging over 10 × 10 km (at nadir). OMI and IASI have nadir resolutions of 13 × 24 km and 12 km (circular footprint), respectively, and have been filtered to remove retrievals with cloud fractions > 0.3 (Fioletov et al., 2011;McLinden et al., 2014;Vinken et al., 2014) and OMI pixels affected by the row anomalies. MODIS, OMI, and IASI provide near daily global coverage, although the row anomalies render portions of the OMI viewing swath unusable. Uncertainty in AOD from MODIS is spatially and temporally variable. It has been estimated as ± (0.05 + 15%) for AOD over land (Levy et al., 2013), and prior research has reported 71% of MODIS Collection 5 retrievals fall within 0.05 ± 20% for AOD relative to AERONET in the study domain (Hyer et al., 2011)."**

P7, L173-174: Give the right uncertainty values i.e. (+-0.05 +15%) and (+-0.05+15-20%).
**Done. See comment above.**

P7, L 184-L 187: Reformulate to explain better how the regridding is done.

**We rephrased as follows:**

**"For the model evaluation, satellite observations for each day are regridded to the WRF-Chem domain by averaging all valid retrievals within: 0.1° and 0.35° for MODIS; 0.125° × 0.18° (along-track/latitudinal × cross-track/longitudinal) and 0.365° × 0.42° for OMI; 0.12° and 0.36° for IASI of each WRF-Chem grid cell centroid, for the 12×12 km and *60×60 km* resolutions, respectively."**

P7, L 190: Standard scores could be shortly explained.

**Done. We rephrased from line 222 as follows:**

**"Model evaluation of gaseous species is performed on a seasonal basis using standard scores (z-scores), which are computed as the difference between the seasonal mean within a grid cell and the seasonal spatial mean, divided by the seasonal spatial standard deviation. The use of standard scores allows comparing spatial patterns of satellite observations and model output in terms of standard deviation units from the mean."**

P8, L206-207: The root mean square difference is not shown in Fig. 1 a)-c).

**We agree. We now refer to Fig. 1 d-f.**

P9, L225-239: This could be explained better. In Murphy and Epstein it is noted that the first term would be the skill if the second and third term were small. The second term is small if for all points F' is linear to P' (conditional bias). The third term gives the overall/mean bias. The fourth term is a correction and should be small.

**We rephrased from line 263 as follows:**

**"The first term in (4) ranges from 0 to 1, is described as the potential skill, and is the square of the spatial correlation coefficient between forecast and reference anomalies to MODIS. It is the skill score achievable if both the conditional bias (second term) and overall bias (third term) were zero, and for most of the variables considered herein (particularly AOD) it contributes to a positive BSS in most calendar months (and seasons). The second term (the conditional bias, > 0), is the square of the difference between the anomaly correlation coefficient and the ratio of standard deviation of the anomalies and is small if for all points $F'$ is linear to $P'$. The third term is referred to as the forecast anomaly bias, and is the ratio of the difference between the mean anomalies of WRF12-remap and the observations relative to WRF60 and the standard deviation of WRF60 anomaly relative to observed values."**

P23, Fig. 2: It would be useful to add the number of cloud-free data points for each season and each of the three datasets (WRF12-remap, WRF60, MODIS).

**We have modified the figure to include the number cloud free grid cells.**

[Figure]

**Figure 2. First line: Number of paired AOD observations at a wavelength (λ) of 550 nm (i.e. simultaneous values as output from WRF-Chem and observed by MODIS) used to perform a t-test designed to evaluate whether the difference computed for each grid cell as WRF60-MODIS differs from that computed as WRF12-remap-MODIS on a seasonal basis (columns show Winter (DJF), Spring (MAM), Summer (JJA) and Fall (SON)). Second line: Results of the t-test. Pixels that have p-values that are significantly different at α=0.10 are indicated in red and have been corrected for multiple testing using a False Discovery Rate approach. The number of observations of cloud-free conditions summed across all days in each season and all grid cells is also reported (black=MODIS, blue=WRF60, red=WRF12-remap).**

**References**

[revised manuscript text omitted]

---

## Author Comment (AC2) · 19 Aug 2016

**Response to review comments on acp-2016-453 from reviewer 2**

**The original comments are provided in black, our response is given below each comment in red.**

**Thank you for the careful reading of our manuscript and your review.**

**Summary**

The present work investigates how an increase in spatial (horizontal) resolution from 60 km to 12 km improves the ability of a mesoscale simulation with the WRF-Chem model to reproduce satellite observations of aerosol optical depth (AOD) and column concentrations of chemical species, and of key meteorological variables (relative to a reanalysis product). The WRF-Chem model is run for the year 2008 on a domain that covers the eastern USA, a region with strong emissions of natural and anthropogenic aerosol precursors.

The motivation of the work is established in the introduction. Simulations, observations, and reanalysis data are subsequently introduced with select relevant details, such as uncertainties in the satellite observations. The authors then provide an overview of the statistical methods used to evaluate model performance relative to observations and reanalysis in a succinct but effective way, which can serve as an introduction to novices to the subject. In the main part of the work the results are presented and analyzed, and the authors carefully quantify and discuss the improvements from an increase in resolution. It is found that a higher simulation resolution improves model fidelity in reproducing observed AOD as well as the ability of the model to identify extreme AOD values. The analysis reveals that the improved performance of the model at higher resolution can in part be attributed to improved agreement in meteorological quantities, in particular boundary layer specific humidity, which contributes to aerosol growth. The model is also shown to better reproduce satellite observations of chemical species at higher resolution. The authors do not fail to identify instances where a higher resolution does not result in an improvement: While model skill (measured by the Brier skill score) in reproducing AOD improves in seven out of twelve months, the model shows improvements in detecting extreme AOD values only in the warm season. In all this, statistical methods are used effectively to quantify model performance. The work addresses an interesting question of general importance in atmospheric modeling - what improvements can be expected from an increase in model resolution? The authors answer this question in exemplary fashion for aerosol properties, chemistry, and meteorology in a mesoscale model. The key insight is that increased spatial (horizontal) resolution clearly improves model performance but is not a panacea. Model aspects other than resolution need attention as well to improve model fidelity. The manuscript is thorough, clear, compelling, very well written, and presents the results with good figures and tables. I recommend publication after attending to the following detailed comments.

**Thank you for your positive assessment. We have addressed your detailed comments below and modified the manuscript accordingly.**

**Detailed comments**

Line 51-52: Please give references for the possible range of values for the direct and indirect aerosol effects.

**Thanks for noting this. We have now added the following reference:**

**Stocker, T. F. a. Q., D. and Plattner, G.-K. and Alexander, L.V. and Allen, S.K. and Bindoff, N.L. and Bréon, F.-M. and Church, J.A. and Cubasch, U. and Emori, S. and Forster, P. and Friedlingstein, P. and Gillett, N. and Gregory, J.M. and Hartmann, D.L. and Jansen, E. and Kirtman, B. and Knutti, R. and Krishna Kumar, K. and Lemke, P. and Marotzke, J. and Masson-Delmotte, V. and Meehl, G.A. and Mokhov, I.I. and Piao,**

S. and Ramaswamy, V. and Randall, D. and Rhein, M. and Rojas, M. and Sabine, C. and Shindell, D. and Talley, L.D. and Vaughan, D.G. and Xie, S.-P. (2013), Summary for Policymakers, in *Climate Change 2013: The Physical Science Basis. Contribution of Working Group I to the Fifth Assessment Report of the Intergovernmental Panel on Climate Change*, edited, pp. 33–115, Cambridge University Press, Cambridge, United Kingdom and New York, NY, USA.

Section 2.1: Aerosol size distributions are generally log-normal, not power law functions. The upper tail of a log-normal distribution does, however, behave like a power law distribution. The discussion here is therefore not incorrect, however, it should be modified to assume a log-normal distribution (also because the aerosol scheme used in the WRF-Chem simulations assumes log-normal distributions).

**Thanks for this comment. We rephrased the paragraph as follows:**

**The relationship between the aerosol size distribution and spectral dependence of AOD is described by a power law function:**

$$\beta(\lambda) = \beta(\lambda_0) \times \lambda^{-\alpha} \quad (1)$$

**where β is the particle extinction coefficient, λ is the wavelength ($\lambda_0 = 1\mu m$) and α is the Ångström exponent (Ångström, 1964) which describes the wavelength dependence of AOD (and is inversely proportional to the average aerosol diameter Dp):**

$$\alpha = \frac{\ln \dfrac{AOD(\lambda_1)}{AOD(\lambda_2)}}{\dfrac{\lambda_2}{\lambda_1}} \quad (2)$$

**The aerosol volume distribution (and thus also its size distribution) can be often described as a multi-lognormal function with *n* modes:**

$$\frac{dV(r)}{d\ln r} = \sum_{i=1}^{n} \frac{C_i}{\sqrt{2\pi}\sigma_i} \exp\left[\frac{-(\ln r - \ln R_i)^2}{2\sigma_i^2}\right] \quad (3)$$

**where $C_i$ is the particle volume concentration in the mode *i*, $R_i$ is the geometric mean radius and $\sigma_i$ is the geometric standard deviation, thus we have:**

$$AOD(\lambda) = \int \frac{3\beta(m, r, \lambda)}{4r} \frac{dV(r)}{d\ln r} d\ln r \, dZ \quad (4)$$

**As indicated in (Schuster, 2006), "the spectral variability of extinction diminishes for particles larger than the incident wavelength", thus fine mode particles contribute more to AOD in the visible (λ~0.5 µm) than at longer wavelengths, whereas coarse mode particles provide a similar AOD both at short and long wavelengths. This is reflected in the Ångström parameter which can be thus used as a proxy for the fine mode fraction or fine mode radius (Schuster, 2006).**

Line 417-418: "Further, the seasonal average spatial patterns of the total columnar concentrations, expressed in terms of z-scores, also exhibit high qualitative agreement with the satellite observations (Fig. S4-S7)."

It is a stretch to write "high qualitative agreement" here. Comparing the OMI, WRF60, and WRF12-remap panels in Fig. S4-S7, my impression is that omitting "high" and leaving "qualitative agreement" is a more accurate assessment.

**We agree. We modified the text accordingly and removed "high".**

Table 1: A factor of 9 is placed in the denominator of the Bonferroni correction, but the data at the different wavelengths, resolutions, and remappings would not seem to be truly independent significance tests. For example, WRF12 and WRF12-remap would seem to be very dependent. Please carefully consider (and justify) whether the factor of 9 should be used or rather omitted.

**We agree that some of the simulations are not really independent as also are the twelve months of the year. The Bonferroni correction can be applied to any set of experiments, either dependent or independent, and aims at providing the most conservative indication of the significance of a statistical test. Therefore we decided to keep the original significance assessment (i.e. with the factor 9) and clarified the explanation of the Bonferroni correction by removing the reference to independence at line 225 as follows:**

**"To assess the significance of ρ while accounting for multiple testing, we apply a Bonferroni correction (Simes, 1986) in which for *m* hypothesis tests, the null hypothesis is rejected if $p \leq \dfrac{\alpha}{m}$ , where *p* is the p-value and α is the confidence level (0.05 is used here)."**

Figure S5: OMI, WRF60, and WRF12-remap panels (or panel titles) are shuffled relative to the panel order in Figures S4, S6, and S7.

**Thanks for pointing this. We have now reordered the panels to be consistent with the other figures on the gas phase evaluation.**

[revised manuscript text omitted]

---

## Referee Report (RR1)

**Responses to reviewers comments and revised version of „Value added by high-resolution regional simulations of climate-relevant aerosol properties" by P. Crippa, R. C. Sullivan, A. Thota, S. C. Pryor**

The responses to the reviews do not address the points made by the reviewers adequately, in particular the differences in meteorological variables at 12 km and 60 km horizontal resolution and the precipitation bias at 60 km resolution need to be understood (see below). Publication can only be recommended after major revisions.

General comment:

For a meaningful comparison of AOD between the simulations at 12 km and 60 km horizontal resolution the differences in meteorological variables and their impact on AOD need to be understood. The annual mean precipitation in the studied region should be around 800 -1200 mm with a standard deviation of about 180 - 260 mm (Groisman and Easterling, 1994). The precipitation of the 60 km simulation in Fig. S3 is significantly below these values in many areas. The reason for the difference between the 12km and 60 km simulations could be the different performance of parameterization at different resolutions or internal variability. The discussion of the cumulus scheme by the authors is very welcome and should be added to the main text. It remains to be checked if the difference between the 12 km and 60 km simulations is also due to internal variability. A 60 km simulation is significantly cheaper than a 12 km simulation. 60 km simulations with varying initial conditions can be used to explore the internal variability and if possible reduce the differences in meteorological variables, in particular reduce the precipitation bias.

Specific comments:

P6, L 152: Effects of the boundary conditions are clearly visible in some of the Figures e.g. Fig. 4, Fig. 6, Figs. S1-S3. It should be mentioned in the text that removing the cells at the boundary does not significantly affect the BSS results or the boundary cells should be excluded from the analysis. Otherwise a reader may be confused whether or not the cells at the boundary are included in the analysis and whether or not they affect the results.

P10, L273-L298: Using the BSS and its decomposition in Murphy and Epstein is useful to investigate which one of two simulations has the higher skill. But it would be interesting and within the scope of the paper to know also the skill of each simulation individually. Therefore it would be useful to compute in addition a BSS for each simulation (WRF60 and WRF12-remap) by using climatological values as the reference.

P17, L477-L485: Because wet scavenging by precipitation is removing most of the aerosol globally (Textor et al., 2006) a short discussion how wet scavenging by precipitation affects the comparison of the two resolutions should be added.

Technical corrections:

P6, L134: The Angstrom exponent alpha is the exponent for (lambda1/lambda2), i.e. (lambda1/lambda2)^-alpha.
P6, L138: The natural logarithm is missing in the denominator.
P6, L141: Only 2pi is below the square root, not sigma_i. Sigma is the standard deviation, not the geometric standard deviation. r is not defined.

P6, L145: The variables in this equation depend on z. z is not defined.
P7, L167: There are words missing before representative.

Reference:

Textor, C., Schulz, M., Guibert, S., Kinne, S., Balkanski, Y., Bauer, S., Berntsen, T., Berglen, T., Boucher, O., Chin, M., Dentener, F., Diehl, T., Easter, R., Feichter, H., Fillmore, D., Ghan, S., Ginoux, P., Gong, S., Grini, A., Hendricks, J., Horowitz, L., Huang, P., Isaksen, I., Iversen, I., Kloster, S., Koch, D., Kirkevåg, A., Kristjansson, J. E., Krol, M., Lauer, A., Lamarque, J. F., Liu, X., Montanaro, V., Myhre, G., Penner, J., Pitari, G., Reddy, S., Seland, Ø., Stier, P., Takemura, T., and Tie, X.: Analysis and quantification of the diversities of aerosol life cycles within AeroCom, Atmos. Chem. Phys., 6, 1777-1813, doi:10.5194/acp-6-1777-2006, 2006.

---

## Referee Report (RR2)

**Response to revised version of „Value added by high-resolution regional simulations of climate-relevant aerosol properties" by P. Crippa, R. C. Sullivan, A. Thota, S. C. Pryor**

The authors show that WRF-Chem with the physical and chemical schemes chosen for their study has substantial biases at 60 km horizontal resolution over eastern North America. The biases depend only weakly on the cumulus scheme or lateral boundary conditions. At 12 km horizontal resolution the biases to re-analysis data, in particular for precipitation, are smaller.

The intended quantification of the value added by enhanced resolution in the description of the drivers of aerosol direct radiative forcing over eastern North America cannot be achieved with the current setup as the bias in precipitation implies a bias in wet scavenging (the most important removal mechanism for aerosol particles, as mentioned in the study) and the bias in boundary layer humidity leads to biases in aerosol water uptake and therefore AOD (which is discussed in the study).

Therefore either the focus of the manuscript needs to be changed to discuss the performance of WRF-Chem at different horizontal resolutions in general or the setup needs to be changed for example by running a simulation with 36 km horizontal resolution. Only then can publication be considered.

---

## Referee Report (RR3)

Response to revised version of „The impact of resolution on meteorological, chemical and aerosol properties in regional simulations with WRF-Chem" by P. Crippa, R. C. Sullivan, A. Thota, S. C. Pryor

The revised manuscript documents the differences in meteorological, chemical and aerosol properties in 1 year WRF-Chem simulations of eastern North America at 12km (WRF12/WRF12-remap) and 60km (WRF60) horizontal resolution and investigates some of the reasons of the different performance between the two resolutions. Among other reasons a dry bias in the specific humidity in the boundary layer and a substantial underestimation of total monthly precipitation in WRF60 are identified as causes for the better performance of WRF12/WRF12-remap over WRF60. This is important for the topic of the study and needs to be mentioned in the abstract. After this minor change the manuscript is recommended for publication.

---

## Author Response (AR2)

**Response to review comments on acp-2016-453 from reviewer 1**

**The original comments are provided in black, our response is given below each comment in red.**

**Thank you for your review. We have addressed the general and specific comments below and modified the manuscript accordingly.**

**In order to address your comments we have:**

1) **Conducted an additional year-long simulation at 60 km resolution using a different cumulus scheme**
2) **Conducted an additional year-long simulation at 60 km resolution using a different set of meteorological lateral boundary conditions.**
3) **Conducted an additional suite of evaluations including inclusion of BSS using MODIS climatology as the reference (we note here that of course as we specified in the manuscript 2008 was chosen as a climatologically representative year).**

The responses to the reviews do not address the points made by the reviewers adequately, in particular the differences in meteorological variables at 12 km and 60 km horizontal resolution and the precipitation bias at 60 km resolution need to be understood (see below). Publication can only be recommended after major revisions.

**General comment:**

For a meaningful comparison of AOD between the simulations at 12 km and 60 km horizontal resolution the differences in meteorological variables and their impact on AOD need to be understood. The annual mean precipitation in the studied region should be around 800 -1200 mm with a standard deviation of about 180 - 260 mm (Groisman and Easterling, 1994). The precipitation of the 60 km simulation in Fig. S3 is significantly below these values in many areas. The reason for the difference between the 12km and 60 km simulations could be the different performance of parameterization at different resolutions or internal variability. The discussion of the cumulus scheme by the authors is very welcome and should be added to the main text. It remains to be checked if the difference between the 12 km and 60 km simulations is also due to internal variability. A 60 km simulation is significantly cheaper than a 12 km simulation. 60 km simulations with varying initial conditions can be used to explore the internal variability and if possible reduce the differences in meteorological variables, in particular reduce the precipitation bias.

**We explored model sensitivity to the cumulus parameterizations by applying the Grell-Freitas cumulus scheme (Grell and Freitas, 2014), which is the next generation of the Grell 3D scheme and has been tested with WRF-Chem, following recommendations of NCAR scientists for our particular case study (personal correspondence with [Saide P., Kumar R., Archer Nicholls S.], 2016). Analysis of precipitation seasonal fields (Figure 1 below) do not present significant differences in magnitude and patterns compared to the original simulations adopting the Grell 3D scheme. As a result, also the BSS of both precipitation and AOD at different wavelengths lead to the same original conclusions on the higher performance of WRF12-remap vs WRF60.**

[Figure]

**Figure 1. Seasonal total precipitation (mm) for MERRA-2 (first row), WRF60 with Grell-Freitas cumulus scheme (second row), WRF60 with meteorological boundary conditions from GFS (third row) and WRF12-remap (fourth row). The Grell 3D cumulus scheme is applied to both WRF60-GFS and WRF12-remap.**

**Modification to the text is as follows:**

**In the introduction (lines 126-146):**
**"Based on the performance evaluation of the WRF-Chem simulations that indicate substantial dry bias in the WRF60 simulations and large seasonality in the value-added by enhanced resolution, we conducted two further year-long simulations at 60 km. In the first we held all other simulation conditions constant but selected a different cumulus parameterization. In the second, we held all simulation conditions constant but employed a different set of lateral boundary conditions for the meteorology. In the context of the precipitation biases reported herein it is worthy of note that discrepancies in simulated precipitation regimes are key challenges in regional modelling (both physical and coupled with chemistry). Although the Grell 3D scheme has been successfully applied in a number of prior analysis wherein the model was applied at resolutions in the range of 1-36 km (e.g. (Grell and Dévényi, 2002;Lowrey and Yang, 2008;Nasrollahi et al., 2012;Sun et al., 2014;Zhang et al., 2016)), the North American Regional Climate Change Assessment Program (NARCCAP) simulations with WRF at 50-km were also dry biased in the study domain (Mearns et al., 2012). Although there have been a number of studies that have sought to evaluate different cumulus schemes over different regions at different resolutions, no definitive recommendation has been made regarding the dependence of model's skill on resolution and cumulus parameterization (Arakawa, 2004;Jankov et al., 2005;Nasrollahi et**

al., 2012;Li et al., 2014). Thus, further research is needed to identify the optimal cumulus scheme for use over North America at coarser resolution. Thus, we performed a sensitivity analysis on the cumulus scheme at 60 km by applying the Grell-Freitas parameterization (Grell and Freitas, 2014), which is the next generation of the Grell 3D scheme."

**In the methods** (lines 202-210):
"As described in detail below, in the WRF60 simulations configured as described in Table 1, simulated precipitation during the summer months exhibits substantial dry bias, and the analysis of value added by enhanced simulation resolution exhibited strong seasonality. We performed a sensitivity analysis to the cumulus scheme, by conducting an additional year-long simulation at 60 km using the Grell-Freitas parameterization (Grell and Freitas, 2014), which is an evolution of Grell 3D that is scale-aware and treats some aspects of aerosol-cloud interactions. We also tested the sensitivity of the simulation results to the meteorological boundary conditions, by repeating the WRF60 simulations using output from the Global Forecast System (GFS) at 0.5° resolution every 6 hours to provide the lateral boundary conditions."

**In the results** (lines 509-519):
"Use of the Grell-Freitas parameterization in the WRF60 simulations did not lead to substantially different magnitude and/or spatial patterns of precipitation compared to WRF60 applied with the Grell 3D scheme, and no improvement in agreement with output from MERRA2. The findings of a negative bias in WRF60 simulations without a corresponding overestimation of AOD may appear counter-intuitive since aerosol concentrations (and thus AOD) are dependent on aerosol residence times and analyses of sixteen global models from the AeroCom project indicate wet scavenging is the dominant removal process for most aerosol species in the study area (Hand et al., 2012;Textor et al., 2006). However, the negative precipitation bias in WRF60 simulations appears to be linked to poor representation of surface moisture availability, boundary layer humidity (Fig. 6), and ultimately aerosol water content (and hence AOD)."

In order to test the internal variability, we drove WRF60 with boundary meteorological conditions from the Global Forecast System (GFS) at 0.5 degree resolution every 6 hours and kept the Grell 3D cumulus scheme. Results from this run also show a systematic under-prediction of precipitation over the domain (Figure 1). Some variability in skill metrics are found for AOD (Figure 2 below), although similar conclusions can be drawn regarding the higher performance of WRF12-remap vs WRF60.
Based on these results, it appears that the skill of WRF-Chem with the physics/dynamics schemes adopted in this work is highly sensitive on the spatial resolution and that the sensitivity of the results to the LBC is relatively small.

[Figure]

**Figure 2. BSS for AOD at 550 nm from WRF-Chem simulations at 60km driven by boundary meteorological conditions from NAM12 (black dots) and GFS (red dots). The BSS is computed according to Equation 5 in the manuscript.**

**Modification to the text is as follows:**

**In the introduction (lines 126-131):**
**"Based on the performance evaluation of the WRF-Chem simulations that indicate substantial dry bias in the WRF60 simulations and large seasonality in the value-added by enhanced resolution, we conducted two further year-long simulations at 60 km. In the first we held all other simulation conditions constant but selected a different cumulus parameterization. In the second, we held all simulation conditions constant but employed a different set of lateral boundary conditions for the meteorology."**

**In the methods (lines 207-210):**
**"We also tested the sensitivity of the simulation results to the meteorological boundary conditions, by repeating the WRF60 simulations using output from the Global Forecast System (GFS) at 0.5° resolution every 6 hours to provide the lateral boundary conditions."**

**In the results (lines 422-424):**
**"Interestingly, BSS for most months (excluding September) are higher for the WRF60 simulations conducted using lateral boundary conditions from NAM12 than GFS."**

**For both issues please also refer to the modified Figure 5.**

[Figure]

**Figure 1. (a-c) Brier Skill Scores (BSS, black dots) for monthly mean AOD by calendar month (1=January) for AOD at 470, 550 and 660 nm. In this analysis of model skill WRF12 output is mapped to the WRF60 grid (WRF12-remap) and BSS are computed using MODIS as the target, WRF60 (driven by NAM12 meteorological boundary conditions) as the reference forecast and WRF12-remap as the forecast. Also shown by the color lines are the contributions of different terms to BSS. In panel c the red dots indicate BSS when the reference forecast is WRF60 driven by GFS meteorological boundary conditions. (d) BSS of monthly mean AOD from WRF60 (green dots) and WRF12-remap (blue dots) relative to MODIS monthly mean climatology during 2000-2014 (reference forecast). Monthly mean AOD from MODIS are used as the target. BSS for WRF12-remap in September is -6.1.**

**Specific comments:**
P6, L 152: Effects of the boundary conditions are clearly visible in some of the Figures e.g. Fig. 4, Fig. 6, Figs. S1-S3. It should be mentioned in the text that removing the cells at the boundary does not significantly affect the BSS results or the boundary cells should be excluded from the analysis. Otherwise a reader may be confused whether or not the cells at the boundary are included in the analysis and whether or not they affect the results.

**We agree. We have now clarified that removing the boundary cells does not affect the BSS results. Noted in the results (from line 412):**
**"Although the effects of the boundary conditions appear in some variables (e.g. in Fig. 4 and Figs. S1-S3), the BSS results do not significantly change even when those cells are removed from the analysis."**

P10, L273-L298: Using the BSS and its decomposition in Murphy and Epstein is useful to investigate which one of two simulations has the higher skill. But it would be interesting and within the scope of the paper to know also the skill of each simulation individually. Therefore it would be useful to compute in addition a BSS for each simulation (WRF60 and WRF12-remap) by using climatological values as the reference.

**Done. We computed BSS for WRF60 and WRF12-remap relative to MODIS climatology over the years 2000-2014. Modification of the text (lines 415-422):**
**"When the BSS is used to assess the skill of each model relative to MODIS AOD climatological mean over the years 2000-2014, WRF12-remap is found to add value relative to the climatology (i.e. BSS >0) during summer months and Nov-Jan whereas BSS for WRF60 is positive from late Fall to early Spring (Fig. 5d). The fact that WRF-Chem does not always outperform the climatology is expected since the model is based on time invariant emissions and skills are assessed relative to a year selected to be representative of the AOD climatology. Mean seasonal AOD from MODIS retrievals over the study region during 2008 lie within ±0.2 standard deviations of the climatology (Crippa et al., 2016)."**

P17, L477-L485: Because wet scavenging by precipitation is removing most of the aerosol globally (Textor et al., 2006) a short discussion how wet scavenging by precipitation affects the comparison of the two resolutions should be added.

**We have now added the following discussion on how wet scavenging impacts aerosol properties. Added text reads (lines 512-519):**
**"The findings of a negative bias in WRF60 simulations without a corresponding overestimation of AOD may appear counter-intuitive since aerosol concentrations (and thus AOD) are dependent on aerosol residence times and analyses of sixteen global models from the AeroCom project indicate wet scavenging is the dominant removal process for most aerosol species in the study area (Hand et al., 2012;Textor et al., 2006). However, the negative precipitation bias in WRF60 simulations appears to be linked to poor representation of surface moisture availability, boundary layer humidity (Fig. 6), and ultimately aerosol water content (and hence AOD)."**

**Technical corrections:**
P6, L134: The Angstrom exponent alpha is the exponent for (lambda1/lambda2), i.e. (lambda1/lambda2)^-alpha.
**Thanks, done.**

P6, L138: The natural logarithm is missing in the denominator.
**Agree, added now. Thanks.**

P6, L141: Only 2pi is below the square root, not sigma_i. Sigma is the standard deviation, not the geometric standard deviation. r is not defined.
**Fixed and added definition of r.**

P6, L145: The variables in this equation depend on z. z is not defined.
**Fixed, thanks.**

P7, L167: There are words missing before representative.
**Added "it is".**

Gas phase concentrations (transformed into z-scores) from WRF12-remap show higher agreement with satellite observations during almost all months, as indicated by the positive BSS (Fig. 7b). However given the limited availability of valid satellite observations (especially during months with low radiation intensity), the BSS are likely only robust for the summer months for all species. Nevertheless, with the exception of $NH_3$ during June, BSS for all months are above or close to zero indicating that on average, the enhanced resolution
simulations do improve the quality of the simulation of the gas phase species even when
remapped to 60 km resolution. Further, the seasonal average spatial patterns of the total
columnar concentrations, expressed in terms of z-scores, also exhibit qualitative agreement
with the satellite observations (Fig. S4-S7).

**4 Concluding remarks**

This analysis is one of the first to quantify the impact of model spatial resolution on the
spatio-temporal variability and magnitude of AOD, and does so using simulations for a full
calendar year. Application of WRF-Chem at two different resolutions (60 km and 12 km)
over eastern North America for a representative year (2008) leads to the following
conclusions:

- Higher resolution simulations add value (i.e. enhance the fidelity of AOD at and near
to the peak in the solar spectrum) relative to a coarser run, although the improvement
in model performance is not uniform in space and time. Brier Skill Scores for the
remapped simulations (i.e. output from simulations conducted at 12 km (WRF12)
then averaged to 60 km, WRF12-remap) are positive for ten of twelve calendar moths,
and for AOD($\lambda$=550 nm) exceed 0.5 for seven of twelve months.

- Spatial correlations of output from WRF12 and WRF12-remap with observations
from MODIS are higher than output from a simulation conducted at 60 km during
most months. For example, in contrast to WRF-Chem simulations at 60 km (WRF60),
simulations conducted at 12 km (WRF12) show positive spatial correlations with
MODIS for all $\lambda$ in all calendar months, and particularly during summer ($\rho$ = 0.5-0.7).

- Output from WRF12 and WRF12-remap exhibit highest accord with MODIS
observations in capturing the frequency, magnitude and location of extreme AOD
values during summer when AOD is typically highest. During May-August WRF12-
remap has *Hit Rates* for identification of extreme AOD of 53-78%.

- At least some of the improvement in the accuracy with which AOD is reproduced in
the higher resolution simulations may be due to improved fidelity of specific humidity
and thus more accurate representation of hygroscopic growth of some aerosol
components.

- Higher-resolution simulations also add value in the representation of other key
meteorological variables such as temperature, boundary layer height and precipitation.

Both spatial patterns and precipitation occurrence are better captured by WRF12-
remap.
- At least some of the improvement in the accuracy with which AOD is reproduced in
the higher resolution simulations may be due to improved fidelity of specific humidity
and thus more accurate representation of hygroscopic growth of some aerosol
components.
- Aerosol concentrations (and thus AOD) are dependent on aerosol residence times, and
thus the source and sink time scales. Analysis of sixteen global models from the
AeroCom project indicate wet scavenging is the dominant sink term for the
predominant aerosol species (sulfate and particulate organic matter; ammonium and
nitrates not evaluated in study) in the study area (Hand et al., 2012;Textor et al., 2006)
Thus, the low precipitation bias in WRF60 simulations should result in a high AOD
bias, but may also lead to poor representation of surface moisture availability,
boundary layer humidity, and ultimately aerosol water content. High model to model
discrepancy has been found in simulating aerosol water uptake (Textor et al., 2006).

[revised manuscript text omitted]
 | 0.350 | 0.361 | 0.387 | 0.281 | 0.532 | 0.649 | 0.775 | 0.674 | 0.333 | 0.399 | 0.535 | 0.284 |

---

## Author Response (AR3)

**Response to review comments on acp-2016-453**

**The original comments are provided in black, our response is given in red.**

The authors show that WRF-Chem with the physical and chemical schemes chosen for their study has substantial biases at 60 km horizontal resolution over eastern North America. The biases depend only weakly on the cumulus scheme or lateral boundary conditions. At 12 km horizontal resolution the biases to re-analysis data, in particular for precipitation, are smaller. The intended quantification of the value added by enhanced resolution in the description of the drivers of aerosol direct radiative forcing over eastern North America cannot be achieved with the current setup as the bias in precipitation implies a bias in wet scavenging (the most important removal mechanism for aerosol particles, as mentioned in the study) and the bias in boundary layer humidity leads to biases in aerosol water uptake and therefore AOD (which is discussed in the study).

Therefore either the focus of the manuscript needs to be changed to discuss the performance of WRF-Chem at different horizontal resolutions in general or the setup needs to be changed for example by running a simulation with 36 km horizontal resolution. Only then can publication be considered.

**The reviewer indicated in his/her review: "the focus of the manuscript needs to be changed to discuss the performance of WRF-Chem at different horizontal resolutions in general or the setup needs to be changed for example by running a simulation with 36 km horizontal resolution."**

**Given that we are already seeking to synthesize 5 sets of year-long simulations in a single manuscript, in response to this comment we have modified the manuscript to include a more balanced description of the performance of WRF-Chem at different horizontal resolutions in terms of the meteorological, gas-phase and aerosol properties (and have changed the title to reflect this refocus). A tracked changes version of the manuscript is attached that shows all of the changes we have made.**

**The impact of resolution on meteorological, chemical and aerosol properties in  regional simulations with WRF-Chem**

P. Crippa[1], R. C. Sullivan[2], A. Thota[3], S. C. Pryor[2,3]

[1]COMET, School of Civil Engineering and Geosciences, Cassie Building, Newcastle University, Newcastle upon Tyne, NE1 7RU, UK

[2]Department of Earth and Atmospheric Sciences, Bradfield Hall, 306 Tower Road, Cornell University, Ithaca, NY 14853, USA

[3]Pervasive Technology Institute, Indiana University, Bloomington, IN 47405, USA

*Correspondence to:* P. Crippa (paola.crippa@ncl.ac.uk), School of Civil Engineering and Geosciences, Cassie Building, Room G15, Telephone: +44 (0)191 208 5041, Newcastle University, Newcastle upon Tyne, NE1 7RU, UK

**Abstract**

~~Despite recent advances in global Earth System Models (ESMs), the current global mean aerosol direct and indirect radiative effects remain uncertain, as does their future role in climate forcing and regional manifestations. Reasons for this uncertainty include the high spatio-temporal variability of aerosol populations. Thus, limited area (regional) models applied at higher resolution over specific regions of interest are generally expected to 'add value', i.e. improve the fidelity of the physical-dynamical-chemical processes that induce extreme events and dictate climate forcing, via more realistic representation of spatio-temporal variability. However, added value is not inevitable, and there remains a need to optimize use of numerical resources, and to quantify the impact on simulation fidelity that derives from increased resolution.added valuetheirtheir performance as a function ofofquantifythe sensitivity ofon spatial resolutionvalue added by enhanced spatial resolutiononsofthe 
[revised manuscript text omitted]

---

## Author Response (AR4)

**Response to review comments on acp-2016-453**

**The original comments are provided in black, our response is given in red.**

The revised manuscript documents the differences in meteorological, chemical and aerosol properties in 1 year WRF-Chem simulations of eastern North America at 12km (WRF12/WRF12-remap) and 60km (WRF60) horizontal resolution and investigates some of the reasons of the different performance between the two resolutions. Among other reasons a dry bias in the specific humidity in the boundary layer and a substantial underestimation of total monthly precipitation in WRF60 are identified as causes for the better performance of WRF12/WRF12-remap over WRF60. This is important for the topic of the study and needs to be mentioned in the abstract. After this minor change the manuscript is recommended for publication.

**As indicated by the reviewer we modified the abstract as follows:**

[revised manuscript text omitted]
 | 0.350 | 0.361 | 0.387 | 0.281 | 0.532 | 0.649 | 0.775 | 0.674 | 0.333 | 0.399 | 0.535 | 0.284 |